# Ecological genomics in the Northern krill uncovers loci for local adaptation across ocean basins

Per Unneberg [1], Mårten Larsson [2,3], Anna Olsson[2], Ola Wallerman[2], Anna Petri[4], Ignas Bunikis[4], Olga Vinnere Pettersson [4], Chiara Papetti [5], Astthor Gislason[6], Henrik Glenner[7,8], Joan E. Cartes[9], Leocadio Blanco-Bercial [10], Elena Eriksen[11], Bettina Meyer [12,13,14] & Andreas Wallberg [2] ✉

Krill are vital as food for many marine animals but also impacted by global warming. To learn how they and other zooplankton may adapt to a warmer world we studied local adaptation in the widespread Northern krill (*Meganyctiphanes norvegica*). We assemble and characterize its large genome and compare genome-scale variation among 74 specimens from the colder Atlantic Ocean and warmer Mediterranean Sea. The 19 Gb genome likely evolved through proliferation of retrotransposons, now targeted for inactivation by extensive DNA methylation, and contains many duplicated genes associated with molting and vision. Analysis of 760 million SNPs indicates extensive homogenizing gene-flow among populations. Nevertheless, we detect signatures of adaptive divergence across hundreds of genes, implicated in photoreception, circadian regulation, reproduction and thermal tolerance, indicating polygenic adaptation to light and temperature. The top gene candidate for ecological adaptation was *nrf-6*, a lipid transporter with a Mediterranean variant that may contribute to early spring reproduction. Such variation could become increasingly important for fitness in Atlantic stocks. Our study underscores the widespread but uneven distribution of adaptive variation, necessitating characterization of genetic variation among natural zooplankton populations to understand their adaptive potential, predict risks and support ocean conservation in the face of climate change.

Climate change is affecting all life on Earth and forcing species to move or adapt[1]. Ocean zooplankton are crucial to maintaining food webs and fisheries but face many challenges including increased temperatures and acidification[2–4]. Many planktonic species are shifting toward higher latitudes[4,5], and continued warming is expected to impact marine communities and ecosystem services[6]. The long-term responses to these changes are unclear, but evolutionary adaptation may be important to sustain populations, particularly when physiological and geographical limits have otherwise been reached[7]. This warrants the need to better understand adaptation in key zooplankton that strongly influence marine ecosystems. Krill (Euphausicea; 86 spp.), or euphausiids, are macrozooplankton crustaceans inhabiting all world oceans. Some species include trillions of individuals and are among the most abundant animals on Earth[8–10]. As grazers of smaller plankton and food for fish and mammals, krill are critical links between primary production and higher trophic levels[3]. However, polar krill of both

hemispheres have recently declined[11,12], while boreal species such as the Northern krill spread into new areas[13], impacting native biodiversity[14,15].

The Northern krill is the largest and most abundant North Atlantic krill species, possibly structured into 3–4 basin-scale gene pools[9,16]. While many krill species are stenothermal and have narrow latitudinal ranges, the Northern krill has unusually broad thermal tolerance and range[17,18]. It occurs across a 2–15 °C temperature gradient (Fig. 1a) and breeds within 5–15 °C[9], much wider thermal envelopes than for example the Antarctic krill *Euphausia superba*, which is constrained within −2.0 °C to +4.0 °C and reproductively challenged already at +1.5 °C[19]. Northern krill from different climates vary in metabolism, nutrition, maturation and timing of reproduction that track local seasonal cycles[17,20], ranging from spawning in late winter–early spring in the Mediterranean Sea to summer in the Atlantic Ocean. These

phenotypic variations could have genetic bases, making *M. norvegica* an attractive model for environmental adaptation.

Zooplankton generally have large populations with considerable genetic variation, suggesting they have high potential to adapt to changing environments through natural selection[21]. Many also exhibit extensive larval dispersal and gene flow, which could either interfere with local adaptation or help introduce adaptive variants[21]. Insights into the genetic basis of adaptation are still scarce for zooplankton[22]. In particular, we lack insight into adaptation in euphausiids, due to their large and repetitive genomes ranging between 11 and 48 Gb (4–16× the human genome)[23], which until recently have hindered genetic analysis. The publication of the Antarctic krill genome marked a major step forward but revealed extensive genetic homogeneity and limited adaptive variation in this circumpolar and panmictic species[24]. A comparative analysis of expressed genes in 20 krill species identified

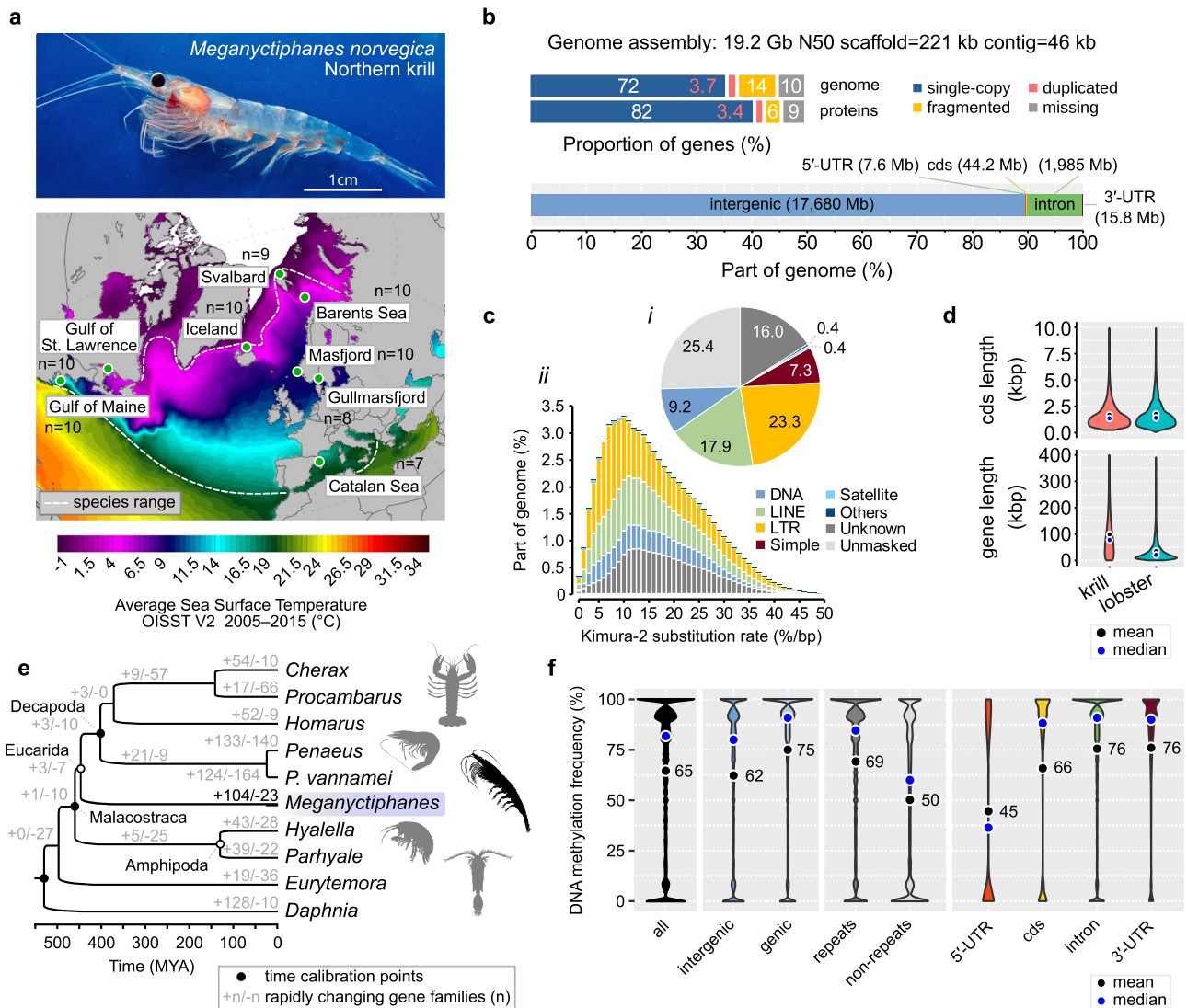

**Fig. 1 | Sampling, genome assembly and genome analyses of the Northern krill *Meganyctiphanes norvegica*. a** Photo: Adult specimen of a Northern krill from the Swedish Gullmarsfjord (approximate scale). Photo by Andreas Wallberg. Map: Atlantic and Mediterranean sample locations indicated in circles (*n* = sample sizes). Sea Surface Temperatures from Climate Reanalyzer[75]. **b** Genome assembly statistics in kilobases (kb), megabases (Mb) or gigabases (Gb). Top: completeness and duplications of BUSCOv5 genes (*n* = 1013) across the genome assembly and protein models. Bottom: sizes of genomic regions (UTR = untranslated exonic regions; cds = coding). **c** The repeat landscape. (i) Proportions of repeat-masked and

unmasked bases (%). (ii) The divergence landscape of interspersed repeats. **d** The coding and full sequence lengths of 7150 single-copy orthologs between *M. norvegica* and the lobster *Homarus americanus*. (gene = exons + introns; cds = coding). **e** Time-calibrated species tree (inferred from 1011 single-copy orthologs; 100% bootstrap support for all nodes) and gene family evolution. **f** The proportion of methylated cytosines across 75 M CpG-dinucleotides in the genome (>10× coverage). Number of interrogated CpG sites: all = 74,890,361; intergenic = 61,170,962; genic = 13,719,271; repeats = 56,713,267; non-repeats = 18,176,966; 5′-UTR = 84,016; cds = 478,953; intron = 13,067,764; 3′-UTR = 88,538.

mechanisms of adaptation but could only reliably interrogate about 1/3 of all genes and focused on cold-adaptation in polar species[25].

Here we used long-read sequencing to assemble the huge genome of the widespread Northern krill. We then performed genome-scale population resequencing with the aim to characterize genetic variation, demographic history, and genetic adaptations to both cold and warm environments at the genome-scale. Insight into adaptive variation can help monitoring processes such as migration and adaptive change, identify hot and cold spots of important functional diversity and support forecasting resilience and risk under continued climate change[26].

## Results

### A highly repeated genome

The Northern krill is diploid and has 19 large metacentric chromosome-pairs that are homomorphic between sexes[27]. We assembled and scaffolded the nuclear and mitochondrial genomes with Nanopore long-reads, Chromium linked-reads and RNA data from mostly a single specimen (Supplementary Table 1, Supplementary Data 1, 2, Supplementary Figs. 1–3). We tracked contiguity, completeness and accuracy by mapping coding transcripts (Supplementary Figs. 4, 5). The finished genome assembly spanned 19.2 Gb ($n = 216,568$ scaffolds/contigs) with ~90% of BUSCO genes present and ~0.5 base-level errors/kb (Fig. 1b, Supplementary Table 2). The genome is GC-poor (%GC = 29.9) and repeat-rich. Low-complexity sequences and simple repeats spanned 15% of the genome (Supplementary Data 3). Using a custom transposable element (TE) library, we found that 74% of the genome is repetitive (Fig. 1c; Supplementary Data 3). While we are unable to classify all TEs, retrotransposons (LINEs+LTRs) outweighed DNA-transposons 4:1, similar to the American lobster or black tiger shrimp[28,29]. The repeatome in the Northern krill exhibits a unimodal pattern of TE divergence (Fig. 1c), whereas the Antarctic krill repeatome is characterized by DNA-transposons and shows signs of several bursts of transposable element proliferation[24]. These observations hint at divergent genome composition and lineage-specific evolution of the gigantic genomes of euphausiids.

### Expansion of cuticular and opsin gene families

We used RNA and comparative data to annotate 25,301 protein-coding genes along with 2283 TEs (mostly expressed retrotransposons), and detected another 14,643 potential yet unannotated genes or TEs (Supplementary Tables 3, 4, Supplementary Data 4, Supplementary Fig. 6). Gene bodies (introns+exons) span 50,276 bp on average and occupy 10% of the genome, while coding sequences cover only 0.22% (Fig. 1b). Orthologous genes between the Northern krill and crustaceans with smaller genomes are 3–8× longer in krill, but have similar amounts of coding sequence (Fig. 1d, Supplementary Fig. 7, Supplementary Table 5, Supplementary Data 5). Compared to the Antarctic krill[24], genes are ~2.5× longer in the Northern krill, suggesting proliferation of retrotransposable elements has produced long and repeated introns (Supplementary Fig. 6). We estimated high synonymous divergence ($dS = 0.46$) between the two species (Supplementary Data 5). Using a decapod molecular clock[30], this divergence suggests they split from a common ancestor ~130 MYA, underscoring separate evolution over long time-scales.

We built a crustacean species tree and analyzed gene family evolution. We found 104 rapidly expanding gene families in the Northern krill ($p < 0.05$; Fig. 1e; Supplementary Fig. 8a, b; Supplementary Data 6), including those related to chitin, cuticular metabolism, regulation of the molting cycle, which are important processes for growth and reproduction in crustaceans. This is notable as renewal of the exoskeleton is unusually frequent and plastic in euphausiids[9,31], and similar expansions were independently detected in the Antarctic krill genome[24]. Moreover, we detected expansions of the opsin gene repertoire, which encodes the light-sensitive receptors in ommatidia.

Fourteen opsins have previously been identified from RNA in *E. superba*[32], which are thought to enable vision under the divergent light conditions experienced throughout its life cycle and vertical migrations[33], while 16 opsins have recently been inferred from *M. norvegica* RNA[34]. We queried our *M. norvegica* gene-set and the *E. superba* RNAs[32] against a curated crustacean opsin dataset. We detected 19 genes in the former species and 15 putative genes in the latter (Supplementary Figs. 9, 10), including new visual middle wavelength-sensitive (MWS) opsins and non-visual arthropsins. All *E. superba* opsins have homologs in the *M. norvegica* genome and all but one previously identified *M. norvegica* transcript can be anchored unambiguously 1:1 to our gene models (Supplementary Fig. 10). Our findings expand the known opsins in both species and suggest that opsin and molting-gene duplications could be common to all euphausiids.

Ancient whole-genome duplication (WGD) may underlie both genome and gene family expansions but is not commonly reported for crustaceans. To test for WGD, we interrogated divergences between gene-paralogs and searched for Hox-gene duplications, but found no supporting evidence (Supplementary Note 1). Instead, the huge Northern krill genome has likely evolved through TE proliferation and numerous small-scale duplications.

### An active DNA methylation system

Epigenetic regulation of the genome may contribute to genome evolution, function and plastic responses to environmental change[35]. CpG-methylation can silence the expression of harmful transposable elements (TEs), paradoxically allowing them to persist and contribute to genome expansion[36]. DNA methylation (DNAm) of both TEs and protein-coding genes is ancestral in Arthropoda, but has frequently been lost[37]. Using DNA from muscle, we characterize the DNAm toolkit and genomic patterns in a euphausiid. The genome had low CpG content ($CpG_{O/E} = 0.53$), indicating DNAm. We found all genes encoding the canonical methyltransferases and genes responsible for repair or demethylation (*dnmt1–3*; *alkB2*; *tet2*; Supplementary Figs. 12–16; Supplementary Data 7), hallmarks of functional DNAm[37]. We scanned the Nanopore-reads for signals of CpG-methylation at 75 million CpG-sites. Overall, DNAm rates were higher in genes than intergenic regions (75% vs. 62% of reads being methylated; Fig. 1f; Supplementary Fig. 17) and positively associated with splice isoform variation ($n = 2.5 \pm 0.05$ isoforms/gene with > 95% methylation rates vs. $1.8 \pm 0.03$ isoforms/gene with <5% methylation rates; Supplementary Fig. 17). In contrast, we observed a lack of methylation in the mitochondrial chromosome (4%). DNAm rates are higher across repeats vs. nonrepeated DNA (69% vs. 50%), and appear to target young retro-transposons with similar LTRs, that may have been recently active (Supplementary Fig. 17f). Gene-body methylation is similar to observations in marbled crayfish[38], while repeat-oriented methylation is reminiscent of the myriapod *Strigamia maritima*[37]. The krill methylome thus spans both genes and repeats, suggesting dual roles in gene regulation and silencing TEs.

### Genome-scale variation is shaped by linked selection and pervasive gene flow

To uncover patterns of genetic variation, we collected 74 Northern krill specimens from eight geographical regions separated by up to 5800 km, covering a wide range of environmental conditions (Fig. 1a, Supplementary Note 2). We resequenced whole genomes to ~3×/specimen (20–30×/population), mapped reads and called 760 million quality-filtered SNPs across 8.4 billion accessible bases (determined with similar filters; Supplementary Fig. 18). We estimated genome-wide nucleotide diversity (π; the average pairwise difference between sequences) and the population mutation rate ($\theta_w$) to be 1.31% and 1.62% per-base, respectively, similar to those inferred from RNA[25] and low compared to marine broadcast spawners such as oysters and sea-squirts[39]. Levels of variation were similar among populations

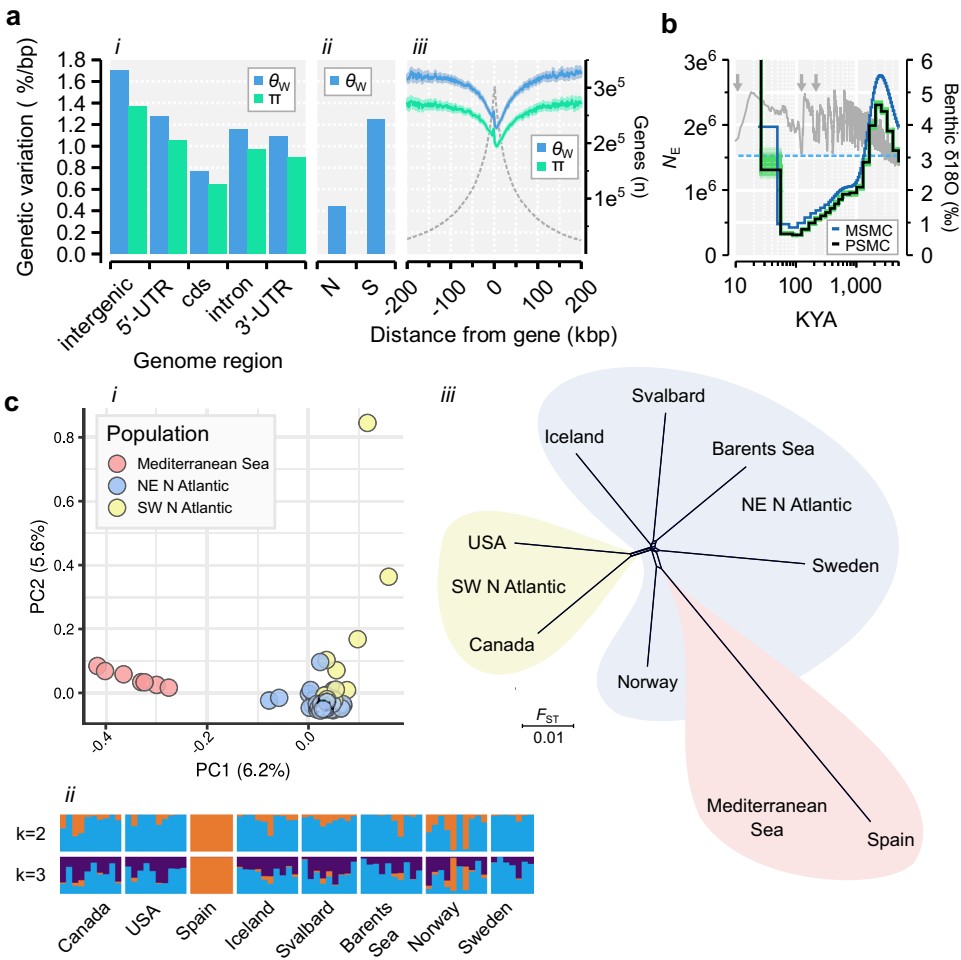

**Fig. 2 | Genome-scale patterns of genetic variation, demographic history and population structure. a** The mean levels of genetic variation (y1-axis) across: (i) the genome; (ii) at nonsynonymous (N) and synonymous sites (S); and (iii) across 1 kb nonoverlapping windows upstream or downstream of putative protein-coding gene bodies (dashed line=number of genes, y2-axis). π is nucleotide diversity and $\theta_W$ is population mutation rate. Shaded regions in *iii* are 95% confidence intervals (200 bootstrap replicates). See Supplementary Data 8 for the number of accessible sites and SNPs. **b** Historical fluctuations in effective population size ($N_E$) inferred using PSMC and MSMC (y1-axis; green=100 PSMC bootstrap replicates; KYA=thousands of years ago). The dashed line is the long-term $N_E$. The LR04 benthic $\delta^{18}O$ isotope (gray line,zf y2-axis) indicates ice volume and sea temperature (arrows=last three interglacials). Number of scaffolds: PSMC = 4911; MSMC = 5176. **c** Interrelationship between samples (n = 74) and populations (n = 8). (i) Main clustering of genetic variation in a principal component analysis (PCA). (ii) Admixture analyses indicating levels of shared ancestry (colors reflect the proportions of major ancestry modes assuming k = 2 or k = 3 hypothetical ancestral populations). For both i and ii 7.35 M downsampled and unlinked SNPs were used. (iii) NeighborNet network of pairwise $F_{ST}$-distances (inferred across 537.5 M SNPs per pair, on average).

(Supplementary Fig. 19A). Assuming mutation-drift equilibrium and a mutation rate from snapping shrimp[30], we inferred the long-term effective population size ($N_C$) to be 1.53 million, far below the expected census population size ($N_C$) of trillions[9]. However, Tajima's D was negative (−0.53), indicating an excess of low-frequency variants compared to expectation under equilibrium, consistent with population expansion shaping genetic variation. We applied the Sequentially Markovian Coalescent to model demographic history from heterozygous sites in the reference specimen[40,41] (~5000 longest scaffolds), suggesting populations expanded halfway through the last glacial period[42] (Fig. 2b). The rate of decay of linkage disequilibrium between SNPs in turn suggested $N_E$ may recently have reached 4–5 million (Supplementary Fig. 19B). We observed ~30% reduction in variation over genes (π = 1.15%), and more so at coding and nonsynonymous sites (Fig. 2a). This effect extended up to 50–100 kb around genes (Fig. 2a), suggesting widespread impact of linked selection on genetic variation.

Population structure was limited but recapitulated geography (the fixation index $F_{ST}$ was ≈0.06 on average; Fig. 2c) and previously

detected mitochondrial gene pools[9]. $F_{ST}$ (relative divergence) increased marginally with geographic distance (Supplementary Figs. 19, 20, Supplementary Data 8). The Mediterranean Sea sample was the most divergent, but the average $d_{XY}$ (absolute divergence) was only 1.04× higher compared to distances among Atlantic populations (1.71% vs. 1.64%). Net synonymous divergence ($D_a$) between Mediterranean and Atlantic stocks was very low (<0.1%), reaffirming they are not reproductively isolated[43]. Moreover, most non-singleton variants were polymorphic in most populations (Supplementary Fig. 21), further indicating extensive gene flow.

## Signatures of ancient adaptive divergence across hundreds of genes

To reveal genomic signatures of adaptation, we partitioned the dataset into two major contrasts: i) Atlantic vs. Mediterranean samples (at/me); and ii) North-Eastern vs. South-Western North Atlantic samples (ea/we; Fig. 3a). For each contrast, we computed pairwise divergence in allele frequencies. While most variation segregated at low differences ($F_{ST(at/me)}$ = 0.056; $F_{ST(ea/we)}$ = 0.017; Fig. 3b; Supplementary

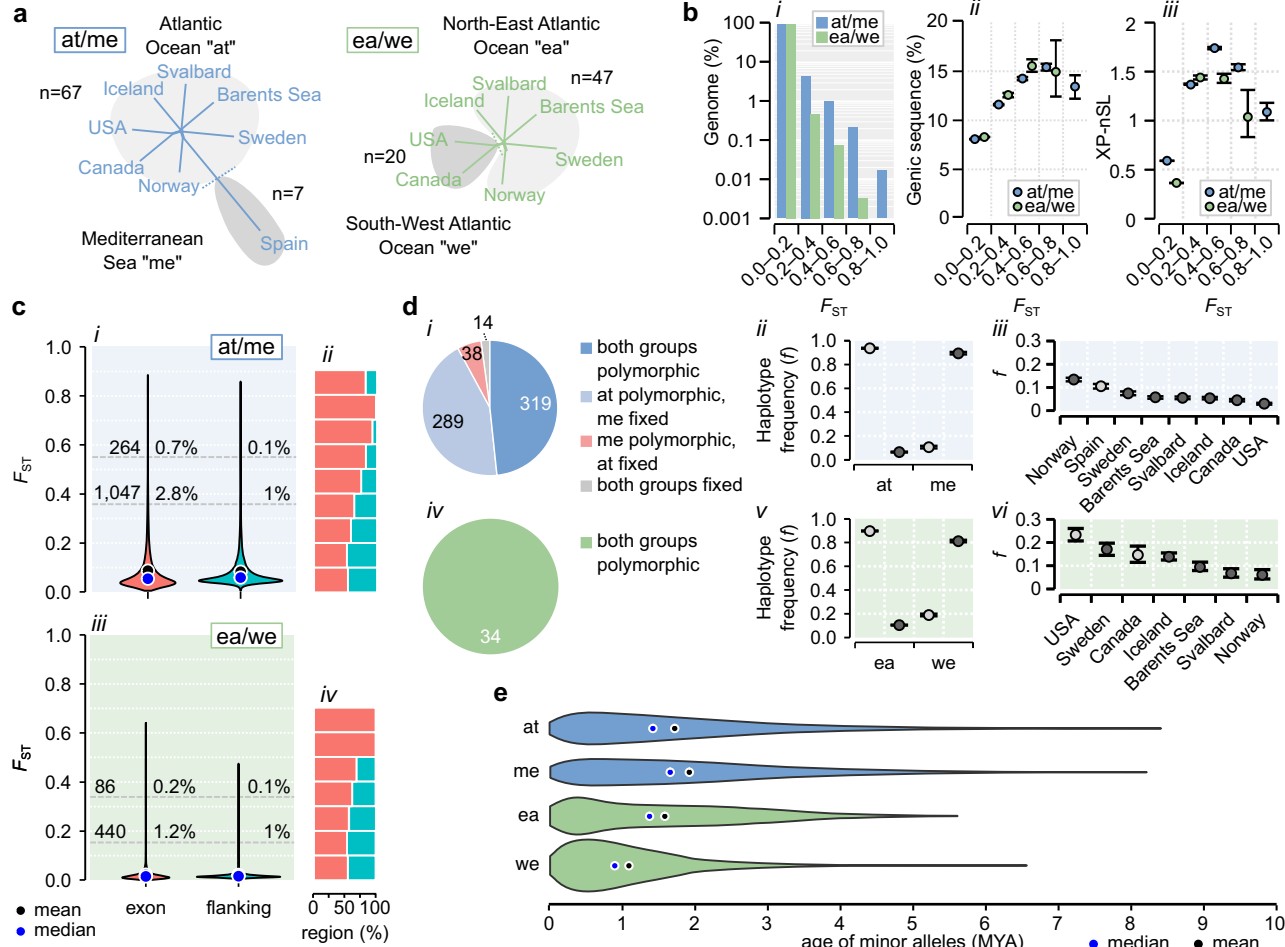

**Fig. 3 | Genetic divergence among Northern krill sampled across the Atlantic Ocean and Mediterranean Sea. a** The two contrasts used to measure divergence. **b** Features of genomic regions at increasing divergence (mean $F_{ST}$ across 1 kb windows, $n = 14$ M). (*i*) The proportion of the genome. (*ii*) Mean proportion of genic sequence (cds+intron+UTRs). (*iii*) The mean absolute extended haplotype statistic | XP-nSL| scores (1 kb windows with ≥50% genic sequence; $n = 1.2$ M). Whiskers in *ii* and *iii* are 95% confidence intervals generated from 2000 bootstrap replicates (two-tailed comparisons). **c** (*i–ii*) Atlantic-Mediterranean contrast: (*i*) The distribution of per-gene $F_{ST}$-values computed across the exons of genes (cds+UTRs; $n = 36,997$) vs. flanking intergenic regions ($n = 31,101$). Gray lines are percentiles of $F_{ST}$ in flanking regions ($F_{ST(1\%)} = 0.36$ and $F_{ST(0.1\%)} = 0.55$). Numbers and proportions of genes are indicated at each percentile. (*ii*) The proportion of genes vs. flanking regions at each $F_{ST}$-level. (*iii–iv*) Eastern-Western contrast: statistics as in *i–ii* ($n = 36,397$ vs. $n = 31,103$). **d** Haplotype distribution for divergent genes (exon-wide $F_{ST} > 0.4$). (*i–iii*) Atlantic-Mediterranean contrast. (*i*) Numbers of genes with shared or private haplotypes ($n = 660$). Mean frequencies of common and rare haplotypes in each group (*ii*) and population (*iii*) (whiskers and 95% confidence intervals as in (**b**)). (*iv–vi*) as in *i–iii* but for 34 genes in the Eastern-Western contrast. **e** Age distributions of minor alleles across (at/me: ~22–25 K SNPs in 660 genes; we/ea: ~2.5 K SNPs in 34 genes).

Fig. 20), we also detected approximately 8× as many highly divergent variants compared to expectations from simulations of neutral drift (Supplementary Fig. 22). Divergent regions ($F_{ST} > 0.4$) spanned ~1% or less of the genome and were about 2× enriched for gene sequences and extended haplotypes, compared to undifferentiated regions ($F_{ST} < 0.2$; Fig. 3b), consistent with gene-centered signatures of selective sweeps[44]. We compared $F_{ST}$ between genes and similarly sized 50 kb flanking regions 50–100 kb away from genes, which may more often evolve neutrally. At high levels of divergence (the top 0.1% most divergent flanking regions), genes outweighed flanking regions by 7× (at/me) or 2× (ea/we), consistent with natural selection driving adaptive divergence across many genes (Fig. 3c; Supplementary Data 8; Supplementary Fig. 23).

We analyzed the geographic distribution of putatively adaptive variation by defining gene-level haplotypes (having at least four diagnostic SNPs with $F_{ST} > 0.5$). At many divergent genes (exon-wide $F_{ST} > 0.4$), both haplotypes were often present in both groups ($n_{at/me} = 319/660$; $n_{ea/we} = 34/34$), indicating widespread standing variation (Fig. 3d). Southern or Scandinavian populations were more

polymorphic than Barents Sea and Svalbard populations (Fig. 3d), which could reflect genetic drift or ongoing selection at the margin of the Arctic species range[45]. We found that Mediterranean haplotypes were comparably common in the Norwegian sample (mean frequency of 13%, Fig. 3d *iii*), while South-Western haplotypes were frequent in the Swedish sample (mean frequency of 17%, Fig. 3d-*vi*), suggesting these Scandinavian stocks in particular could contain genetic material that is otherwise rare at high latitudes. We estimated the ages of minor alleles on the divergent haplotypes to learn for how long haplotypes may have been segregating in the species. We first estimated a genome-average recombination rate ($r = 0.32$ cM/Mb, $n = 652$ scaffolds) and then applied the Genealogical Estimation of Variant Age (GEVA)[46]. This coalescent method infers the time to the most recent common ancestor using mutation and recombination rates, without requiring a priori assumptions about demographic history. Most variation originated over 1 MYA, predating multiple glacial cycles, and adaptive variation segregating between Atlantic and Mediterranean populations may predate that segregating in the Atlantic Ocean (Fig. 3e).

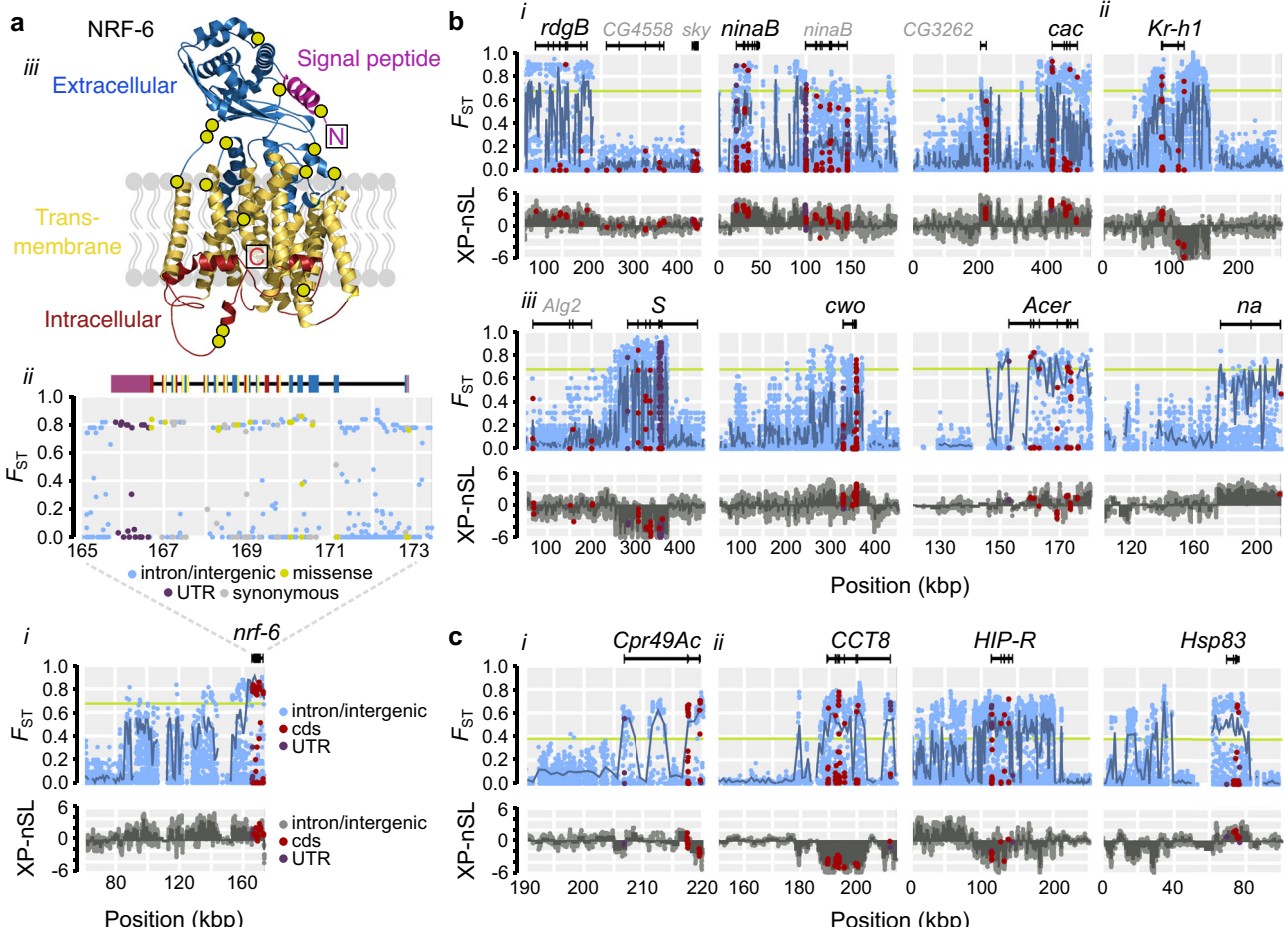

**Fig. 4 | Adaptive divergence and candidates for local adaptation. a** High divergence between Atlantic and Mediterranean samples across the *nrf-6* gene. (*i*) Top: per-SNP $F_{ST}$ along the locus (green line is the 0.1% $F_{ST}$-percentile for SNPs at $F_{ST(0.1\%)} = 0.675$; dark-blue line=$F_{ST}$ for 1 kb windows; black=gene model). Bottom: per-SNP XP-nSL (dark-gray bars=mean window-based XP-nSL). Positive values imply a selective sweep in the Mediterranean sample; negative values mark the Atlantic Ocean sample. (*ii*) Magnified view of *nrf-6*. Exons colored by UTR (purple) or NRF-6 protein topology. (*iii*) Modeled protein structure and topology (including signaling peptide; boxes=N/C terminals; yellow circles=nonsynonymous/missense variants). **b** Examples of highly differentiated genes between Atlantic and Mediterranean samples ($F_{ST(0.1\%)} = 0.675$). (*i*) Photoreception: *retinal degeneration B* (*rdgB*), *neither inactivation nor afterpotential B* (*ninaB*) and *cacophony* (*cac*). (*ii*) Heterochronic development: *Kruppel homolog 1* (*Kr-h1*). (*iii*) Circadian regulation: *Star* (*S*), *clockwork orange* (*cwo*), *Angiotensin-converting enzyme-related* (*Acer*) and *narrow* (*na*). Statistics as in (**a**). **c** Examples of highly differentiated genes between North-Eastern and South-Western North Atlantic samples ($F_{ST(0.1\%)} = 0.38$). (*i*) The top gene in the Atlantic comparison was *Cuticular protein 49Ac* (*Cpr49Ac*) (*ii*) Three chaperon genes implied thermal tolerance: *Chaperonin containing TCP1 subunit 8* (*CCT8*), *Hsc/Hsp70-interacting protein related* (*HIP-R*) and *Heat shock protein 83* (*Hsp83*).

## Candidate genes for ecological adaptation associated with eco-physiological functions

For each contrast, we ranked all genes by exon-wide $F_{ST}$. The top gene between Atlantic and Mediterranean krill is *nose resistant to fluoxetine protein 6* (*nrf-6*) (Supplementary Data 8), encoding a membrane protein facilitating lipid transport. It is located within a high-divergence region and the Mediterranean samples are fixed for a *nrf-6* haplotype with low frequency among Atlantic samples (6%). Exon-wide $F_{ST}$ is > 0.8 and high XP-nSL scores indicate loss of variation consistent with a selective sweep among Mediterranean samples (Fig. 4a). We also detect accelerated protein evolution on the Mediterranean haplotype (*dN/dS*: 0.54 vs. 0.31; Supplementary Fig. 24), consistent with positive selection. We predicted the protein topology and detected 1.7× enrichment of missense variants in its extracellular part compared to synonymous variants (9/16 vs. 5/15; Fig. 4a; Supplementary Fig. 24; Supplementary Data 9), which may alter its function. This is noteworthy as *nrf-6* is important for yolk transport into eggs in worm[47] and ovary development and oogenesis in fly[48], and is overexpressed in the ovaries of sexually precocious crabs[49]. In the Northern krill, Mediterranean stocks depend on a short spring phytoplankton bloom to accumulate lipid stores and trigger vitellogenesis and reproduction in early spring[9,50]. The Mediterranean *nrf-6* variant could contribute to advanced reproductive timing, making it a strong candidate for local adaptation.

We next queried the ranked lists for enriched gene ontologies using fly homologs. Atlantic-Mediterranean divergence was enriched for genes involved in oogenesis, muscle function, phototransduction and eye development, regulation of circadian rhythm and heterochronic development (Supplementary Fig. 25; Supplementary Data 10). These include homologs for *ninaB* that synthesize visual pigment[51], *S* that regulate sleep/wake cycles and the transcription factor *Kr-h1* that acts in the juvenile hormone signaling pathway to govern vitellogenesis and reproduction in arthropods[52] (Fig. 4b). Overall, 30 genes in the 0.1%-percentile ($F_{ST} = 0.55–0.72$; $n = 264$) were associated with vision-related ontologies, 1.5× more than expected by chance ($P = 0.019$), and 112 genes in the 1%-percentile ($F_{ST} = 0.36–0.72$; $n = 1024$), including four MWS opsins (Supplementary Data 10). Photoreception candidates often involved paralogs (Supplementary Fig. 26) and belonged to rapidly expanded gene families, which were otherwise underrepresented in the 1%-percentile (2.98× vs. 0.6×; Supplementary Data 10).

Between the two Atlantic basins, the most divergent gene encodes a larval cuticular protein (homolog of *Cpr49Ac*; Fig. 4c), while this contrast was enriched for signaling ontologies (Supplementary Figs. 25). Among the top 20 genes we identified three heat-shock chaperones/co-chaperones, including *CCT8* (Fig. 4c), that fold proteins and promote proteostasis under thermal stress. Heat-shock proteins contribute to protective thermal tolerance in krill and many other species[53]. Chaperonins (CCT-n) influence cold shock response in crustaceans and other eukaryotes[54]. Multiple CCT genes were linked to cold-adaptation in krill[25] and *CCT8* evolution was implicated in freeze-tolerance in Amur sleeper fish[55]. North-Eastern krill have a sweep-like signature of extended *CCT8* haplotypes, suggesting selection for cold tolerance in these stocks.

## Discussion

Here, we provide insight into the evolutionary history of the Northern krill. Our genome assembly reveals a highly methylated repeatome and many expanded gene families. Paralogous genes may have arisen from ectopic recombination between nonhomologous repeated loci, which is more likely to occur when genomes accumulate transposable elements[56]. The krill genome appears to continuously have evolved new genes, including those involved in molting and vision.

Molting is a crucial process in krill, being interlinked with growth and reproduction and a means to reduce load and drag from parasites and epibionts[57,58]. It is controlled by environmental cues including light[9,59]. Some of the top gene candidates for local adaptation belong to expanded gene families, including two *ninaB* paralogs and four *ninaE* paralogs that synthesize visual pigment or encode MWS opsins. We detected elevated divergence in fifteen genes encoding cuticular proteins. Expanded cuticular and opsin gene families and functionally diverged paralogs may have enabled development and photoreception under diverse conditions in ancestral krill and provide substrates for adaptation today[60].

We characterized genome variation in *M. norvegica* and found moderate levels of diversity, consistent with observations in other krill[24,25]. This may partly be explained by linked selection and recent population bottlenecks purging variation[61]. Likewise, our estimate of long-term effective population size is much smaller than census populations but assumes mutation-drift equilibrium, which krill may rarely have time to reach before evolutionary and demographic processes influence variation[61]. Euphausiid life history traits, including mating where a single male can fertilize many eggs by plugging the female's genital opening with a spermatophore[9,62], may additionally contribute to skewed reproductive success and limit $N_E$ and genetic variation[61]. Further research is needed to understand the size and determinants of effective populations in krill.

We studied divergence among populations to identify genes that may contribute to ecological adaptation. We detected small-to-moderate shifts in allele frequencies in many functionally related genes, consistent with signatures of polygenic adaptation[63]. Evolve-and-resequence experiments in copepods have similarly found polygenic signals of adaptation to temperature and acidity in laboratory conditions[64]. Genetic adaptation in zooplankton may commonly involve numerous loci, warranting genome-scale assays to map adaptive variation. Many of our candidate genes have roles in photoreception, circadian rhythm, and oogenesis − ecophysiological functions also implied in adaptation in other widespread pelagic species, such as the Atlantic herring[65]. The variants may help krill respond to light, temperature and resources in different environments and latitudes.

Photoreception varies among *M. norvegica* populations: krill from turbid waters around the Gulf of Maine are more light sensitive than those from clearer waters[66]. Water clarity and light penetration influence behavior in *M. norvegica*. To avoid predators, stocks prefer deeper depths in clearer Norwegian fjords[67], while the deepest daytime depths (400–800 m) are known from the oligotrophic Mediterranean

Sea[9]. Moreover, light sensitivities in the North Pacific krill *E. pacifica* are tuned to local conditions. Individuals inhabiting shallow green water in the Saanich Inlet are more sensitive to green light compared to those from the deeper blue water of the San Diego Trough[62]. At least thirty genes involved in eye function diverge strongly between Atlantic and Mediterranean krill (including 4 *ninaE*/MWS paralogs), and another six genes segregate across the Atlantic Ocean, suggesting heritable variation could contribute to these phenotypes. Eye traits are generally fast-evolving among krill species and are associated with ecological niche[68]. Our candidates could help reconstruct the genomic architecture of vision and behavior in krill.

Seasonal and daily cycles of ambient light are central to zooplankton and influence diapause, vertical migrations and entrainment of endogenous circadian clocks that control the daily rhythm of physiological processes[69]. *M. norvegica*, along with other krill species such as *E. superba* and *Thysanoessa inermis*, exhibit endogenous circadian rhythms (ECR) that cycle at rates faster than 24 hours in the absence of light[70,71], or respond to minute irradiance. This may reflect a shared light-sensitive circadian toolkit common to all krill or adaptations to photoperiodic variability at high-latitudes where the sun is either below or above the horizon for extended periods[71,72], although the genetic mechanisms of these adaptations are unknown. *M. norvegica* expresses circadian clock genes[73] and we find that genes likely involved in regulating its circadian clock diverge across its geographic range, including homologs of *narrow abdomen*, *glass* and *clockwork orange*. Using *M. norvegica* as a model, functional assessments of local variants could illuminate how biological clocks are set in different environments.

Marine ecosystems are changing at unprecedented rates, driving redistribution of organisms and impacting food webs[5,15]. Antarctic krill is already declining, which could severely impact the Antarctic ecosystem[11,14]. The Northern krill could be declining around Iceland[74], but increasing in the Barents Sea[13]. Where will krill thrive in the future? We found many variants that could help krill adapt to new or changing environments, many of which are widely but unevenly distributed, old and possibly maintained by long-term balancing selection under slowly fluctuating conditions. These adaptive variants could be important for coping with rapid climate change, and Scandinavian stocks in particular may serve as sources of genetic diversity otherwise associated with warmer locations. The next frontier for the Northern krill is the Arctic Ocean[13,15]. Standing variants supporting physiological processes under darker or colder conditions may help establish it there. The Mediterranean Sea population is the most divergent, and close to the species' southern limit. This population appears to lack variation at many adaptive loci, which might limit its evolutionary potential, although our analysis is limited by a small sample size.

With the exception of the Antarctic krill[8], long-term monitoring of krill abundance is typically performed or reported in aggregate[12], obscuring how individual species fare under climate change. Our results suggest that krill may commonly be genetically fine-tuned to their environments, while previous research underscore limited thermal tolerances in many species[18,19]. Genetic adaptation could be the major process determining whether krill will persist or perish. The many candidate genes reported here can be used as biomarkers to diagnose and monitor change of adaptive variation in response to changing conditions, including *CCT8*, *HIP-R* and *Hsp83* associated with thermal stress response and *nrf-6*, *Kr-h1* and *Hr3* involved in reproductive physiology, as well as those potentially coordinating photoperiodic regulation. These markers could also be used to monitor other species, in order to better forecast the future distributions of krill and estimate risks of the great many species that depend on them.

## Methods
### Biological materials
Adult specimens of the Northern krill (*Meganyctiphanes norvegica*) were collected from eight geographical regions across the North

Atlantic Ocean and the Mediterranean Sea by different collaborators and co-authors (Supplementary Table 1; Fig. 1a; Supplementary Fig. 19C). Environmental characteristics and sampling at each location is provided in Supplementary Note 2, while Sea Surface Temperatures in Fig. 1a are from Climate Reanalyzer using NOAA OI SST V2 High Resolution Dataset data provided by the NOAA PSL, Boulder, Colorado, USA[75,76]. Sample details including sampling coordinates and sequencing yields are given in Supplementary Data 1.

**Processing of samples.** Live specimens were either preserved in 95-99% EtOH or in RNA*later*™ (Invitrogen™) to facilitate DNA and RNA recovery. Some specimens were split behind the carapace so that the cephalothorax and abdomen could be stored separately in either buffer. For most samples, the sex was determined by inspecting petasma or thelycum for male or female characteristics using a stereo microscope. This was either done at the time of collection or after preservation in EtOH. Altogether, 36 samples were determined to be males, 32 to be females. Seven samples were not sexed because the relevant tissue had been removed or damaged in processing.

A single female specimen (Sample IDs "K20" or "swe_1"; ~3 cm long) collected in the Gullmarsfjord in Sweden was used to produce all primary long-read and linked-read DNA data for the main de novo genome assembly, as well as most novel short-read and long-read RNA/cDNA resources used for scaffolding and annotation of the genome sequence. In order to reduce the risk of contamination from food-particles, this specimen and others sampled at the same location were kept in aquaria with filtered running deep water at ambient temperatures (10 °C) at the Kristineberg Marine Research Station without access to food for 24–48 h before they were preserved. Sequences from other specimens were used to assist in specific tasks for the genome assembly (Supplementary Table 1; Supplementary Fig. 2), as well as to produce a comprehensive population dataset to map genetic variation.

### Overview of library preparation & sequencing

For the genome assembly and annotation, we produced DNA and RNA/cDNA data using Oxford Nanopore Technologies (ONT) MinION (hereafter: MinION) and PromethION long-read sequencing (hereafter: PromethION). Linked-read DNA data was produced using 10x Genomics Chromium sequencing (hereafter: 10x) (Cat#120257/58/61/62). RNA was also sequenced using Illumina RNA-seq. Low-coverage whole-genome population-scale resequencing (WGS) was carried out using Nextera DNA Flex libraries (Illumina Cat#20018705) or Illumina DNA PCR-Free libraries (Illumina Cat #20041795). All DNA/RNA extractions and the MinION sequencing were performed at the Dept. of Medical Biochemistry and Microbiology at Uppsala University. All other library preparation and sequencing was carried out by the National Genomics Infrastructure (NGI) at Science for Life Laboratory (SciLifeLab), Sweden. Extractions were quantified for purity, yield and fragment lengths using the Thermo Scientific NanoDrop and Qubit systems, standard 1% agarose gels, and Agilent TapeStation and Femto Pulse systems, respectively. Summary reports with basic sequence statistics were produced using PycoQC for long-reads[77] and MultiQC for short-reads[78] with the high-throughput FastQC backend (Simon Andrews, Babraham Institute in Cambridge), respectively, according to standard procedure. See Supplementary Data 2 for additional kit catalog IDs.

**High-molecular weight DNA extraction for long-read and linked-read libraries.** We produced DNA data using PromethION and 10x sequencing of a single female specimen for the genome assembly. We dissected five abdominal segments for all ethanol-preserved tail muscle tissue and extracted each segment individually (K20-1 through K20-5). Each piece was snap frozen in liquid nitrogen and grinded using a pestle and mortar. High-molecular weight (HMW) DNA was extracted using the Genomic-tip 100/G Midi kit (Qiagen) according to the manufacturer's protocol for tissues. Briefly, the pieces of tissue were lysed

in presence of Protease (Qiagen) and RNAse A (Qiagen) to degrade proteins and RNA, respectively. The DNA was bound to the Genomic-tip resin, purified from RNA, protein and other contaminants, eluted and desalted using isopropanol precipitation. The DNA extractions were finally eluted in 70 μL of TE buffer (Qiagen AE buffer). The extraction yielded approximately 30 μg DNA (Qubit), with a peak of DNA fragment lengths at about 130 to 180 kbp, and with similar yields and qualities across the five abdominal segments (Supplementary Fig. 1A, B).

**PromethION long-read sequencing.** The ONT SQK-LSK109 Ligation Sequencing Kit was used by NGI to prepare DNA libraries using all five extractions (K20 1–5; 90% of DNA was consumed), which were then sequenced across ten PromethION FLO-PRO002 R9.4 flow cells. The DNA was sheared to 20 or 30 kb fragments with a Diagenode Diagnostics Megaruptor before preparing most remaining libraries to find an optimal balance and tradeoff between read lengths and yields (Supplementary Fig. 1C). One HMW library was made with the Megaruptor set to shear to 75 kb, followed by size selection for fragments > 15 kb using the Sage Science BluePippin system. Library preparation for nanopore sequencing was done using the LSK-109 kit, with the following changes to the original protocol: i) End-prep / FFPE incubation times at 20 and 65 degrees were prolonged up to 10 or 15 min, and ii) adapter ligation time was prolonged to up to 55 min. Sequencing was performed on a PromethION beta machine, using R.9.4 flow cells. In order to increase the overall yield, we reused the flow cells. We first washed them using standard wash kit buffers and then performed nuclease flushes to digest old libraries, before applying new DNA. Two flushed flow cells were run with cDNA libraries instead of DNA (see below).

Sequencing was performed using ONT MinKNOW v3.1.18. Initial basecalls were carried out using MinKNOW/Albacore but were updated with calls using ONT Guppy v2.3.7 and the more accurate "flip-flop" algorithm (the dna_r9.4.1_450bps_flipflop_prom model)[79]. The updated basecalls produced 593.8 Gb of DNA data (13% produced on flushed cells), representing 13% more data than the original basecalls. The Guppy-recall took 10 days of GPU-compute time across four NVIDIA 1080 GPUs. We sequenced 84.4 M reads with overall $N_{50}$ = 12 kb. About 57% of sequences had mean base qualities (Q-scores) above 10 using flip-flop (Supplementary Data 2), compared to 16% in the original call. Assuming a genome size of 18.6 Gbp[23], the PromethION data corresponded to 32× coverage across the genome, with ~20× coverage of reads 10 kb or longer (Supplementary Data 2; Supplementary Fig. 1D).

**10X linked-read sequencing.** Four separate 10x GEM libraries were made from the K20-5 HMW DNA extraction by NGI following the manufacturer's recommended protocol and had mean library fragment lengths of 530–540 bp. Paired-end reads (2×150 bp) were sequenced on a single Illumina NovaSeq 6000 S4 lane, generating 897 Gb of data across 2.97 B read-pairs (~46× coverage; 12.6% marked as duplicates; %GC = 34.7). The standard LongRanger (v2.2.2; 10x Genomics) "basic" command was used to extract the 16 bp barcodes from the forward/R1 reads for each library individually. About 3.5 M of the 4.8 M barcodes (4M-with-alts-february-2016.txt; 10x Genomics) were found to be associated with at least one read in each library and ~96% of all sequences were barcoded. We rewrote the "BX:Z" style barcodes of each of the four libraries using a unique suffix ("_1" through "_4") in order to keep them separate in downstream analyses and to de-interleave the FASTQ files.

**RNA extraction from multiple tissues for long-read and short-read RNA libraries.** We produced RNA data using PromethION/minION cDNA-sequencing and Illumina short-read RNA-seq using the K20 specimen for gene and genome annotation. Six libraries were prepared using RNA from small pieces of tissue preserved in RNAlater:

i) head/brain; ii) dorsal/caudal muscle tissue on cephalothorax; iii) thoracic exopodite legs (distal segments of the periopods); iv) ventral soft tissue of cephalothorax including gills; v) hepatopancreas (the crustacean digestive gland); and vi) soft-tissue surrounding it. RNA was extracted using the RNeasy Lipid Tissue Mini Kit (Qiagen) according to the standard protocol, using ceramic beads and a BioSpec Products Mini-Beadbeater for tissue homogenization, and eluted in 30 μL RNAse-free water. This yielded RNA extractions with concentrations between 50 and 240 ng/μL and with RIN-values between 9.1 and 10. We used both raw RNA-seq reads and Trinity assembled transcripts[80] in downstream analyses.

**Short-read RNA sequencing.** NGI produced six libraries from the six K20 RNA extractions using Illumina TruSeq Stranded mRNA Library Prep kit with polyA selection (Illumina Cat# 20020594/5, manufacturers' protocol #1000000040498), which were sequenced as 2 × 150 bp paired-end reads on one Illumina NovaSeq 6000 SP lane. This generated 459 M read-pairs that were demultiplexed by NGI (Supplementary Data 2).

**Long-read cDNA sequencing.** One barcoded and amplified Nanopore cDNA library was produced from each of the six K20 RNA extractions, using the PCR-based ONT Strand-switching protocol with poly-A selection and the LSK109 ligation kit to add barcodes and adapters. We performed cDNA sequencing on PromethION (ONT MinKNOW v3.1.18) using two nuclease flushed flow cells and one new FLO-MIN106D R9.4.1 MinION flow cell (3 M reads) and the standard loading kit and buffers for each instrument. The data was basecalled using ONT Guppy v4.4.1 and HAC (high accuracy) model (dna_r9.4.1_450bps_hac_prom.cfg). The two PromethION runs generated 8.99 M and 3.91 M reads (10.1 + 5.2 Gb of cDNA data), with median Q-scores of 10.5 and 10.7 and median lengths of 962 bp and 1060 bp, respectively. The MinION run produced 3.0 M reads (3.5 Gb of cDNA data) with median Q-score of 12.6 and median lengths of 915 bp.

**Salt-isopropanol DNA extraction of multiple specimens for population-scale whole-genome resequencing.** To produce the population dataset, we extracted DNA from 75 *M. norvegica* specimens using a salt-isopropanol precipitation protocol. We extracted DNA from abdominal muscle tissue from 1–2 muscle segments of krill samples. Samples were selected based on geographic origin and quality of preservation. We sequenced 6–10 specimens per population. We used a column-free salt-isopropanol precipitation protocol to extract DNA. Briefly, tissues were lysed over night in ATL Lysis buffer (Qiagen) together with Proteinase K (Qiagen) over night, followed by RNAse A (Qiagen) treatment, protein separation using Protein Precipitation Solution (Qiagen) and precipitation and washing of the DNA using isopropanol and ethanol, respectively. The washed DNA pellet was finally eluted in 70 μL MilliQ-water. The extractions produced about 7 μg of DNA on average per specimen (Qubit). The protocol is available as a separate document in this publication and at the online platform protocols.io:

- https://www.protocols.io/workspaces/wallberg.lab.imbim.uu/resources/isopropanol-hmw-dna-extraction-method

**Short-read DNA libraries and resequencing.** Barcoded Nextera DNA Flex or Illumina DNA PCR-Free libraries (Illumina) were prepared from 75 DNA extractions (74 specimens for resequencing + the reference specimen) by NGI/SciLifeLab according to the manufacturer's instructions. Both protocols use a bead-linked transposome to fragment DNA and incorporate adapters in a single step. About 350 ng of DNA was used as input per Nextera library and 300 ng per PCR-Free library. We aimed for ~3× depth of coverage per specimen and sequenced the libraries across six Illumina NovaSeq 6000 S4 lanes, generating 15.1 B 2×150 bp read-pairs with valid library barcodes (18%

or 23% of reads marked as duplicates with Nextera or DNA PCR-Free, respectively; %GC = 32.9; 88% of reads with Q-scores above 30). One library ("sva_1" from the Svalbard area) failed to sequence (0.03× coverage; Supplementary Data 1) and was discarded from population genetic analyses (hence 74 samples were carried forward).

### A preliminary mitochondrial assembly

A minION test run of specimen K4 from the Gullmarsfjord produced 1.5 Gbp of long-read across 270,185 reads (N50 = 8229 bp; < 0.1× across the genome) ahead of the other sequencing runs. We downloaded mitochondrial *M. norvegica* template accessions (16 S = AY744910; COI = FJ581747; CYTB = AF149775; NADH = AF149775; tRNA_ser=AF149775; tRNA_leu_16S.AF149775) and used BLASTN 2.2.31+ to identify 235 putatively mitochondrial reads shorter than 16 kbp that aligned to any of the templates. These reads were assembled in Canu v1.7[81] with "genomeSize=17k" that produced a 24.8 kbp contig. We blasted the read pool against this contig and collected additional reads (n = 520 in total) and performed a second round of assembly with Canu, generating a new 24.802 bp contig from 83 aligned reads (37× depth of coverage) that the assembler suggested was circular. Manual inspection of the sequence indicated that it was the full mitochondrial chromosome partly repeating itself and we circularized the sequence by hand, producing a 17,360 bp long sequence. We next mapped the long-reads to the mitochondrial sequence with minimap2 v2.17-r941[82] (109× mean depth of coverage) and polished it with Nanopolish v0.11.3[83] using the "variants" and "vcf2fasta" subcommands. We took the high coverage as an indicator of high mitochondrial:nuclear ratio in the extracted muscle tissue. Finally, we mapped RNA-seq data from the published TI787-MN-F specimen to the polished sequence using Bowtie2 v2.3.4.1[84] and performed one round of polishing with Pilon v1.22[85] with the "--fix indels,gaps,local" setting. The resulting mitochondrial sequence was 17,508 bp with %GC = 28. Automated annotation was performed with the online MITOS2 tool[86]. We used this sequence to screen out potential mitochondrial long-reads from the K20 reference datasets ahead of genome assembly and used a similar approach to assemble the mitochondrial chromosome from this specimen (see mitochondrial re-assembly below).

### Pre-processing of long-read DNA data

We used Porechop v0.2.4[79] to trim the LSK109/NSK007 top and bottom adapters from the PromethION long-reads. Reads with internal adapters can be chimeras and either be split or removed. We first ran the program without splitting potentially chimeric reads:

    porechop -t 20 --no_split --adapter_threshold 60 --end_threshold 70 --end_size 200 --in fastq.gz --out fastq.trimmed.gz

We then repeated the scan without "--no_split" to note which reads had middle adapters (n = 50,368; 0.06% of all reads). We next mapped all reads against our preliminary mitochondrial sequence with minimap2 v2.17-r941 and found that 382 K reads mapped to this chromosome for a mean depth coverage of 33,662× (vs about 32× across the nuclear genome). We filtered out both the putative chimeric and mitochondrial long-reads for the reference PromethION FASTQ readpool, to reduce the risk of downstream interference in the main genome assembly.

### RNA processing and transcriptome assembly

**Long-read cDNA processing.** The barcoded ONT PromethION and minION cDNA reads were de-multiplexed with ONT guppy_barcoder v4.4.1 and then processed with ONT Pychopper v2.5.0 (https://github.com/nanoporetech/pychopper) to find, orient and trim full-length cDNA reads using the SSP/NVP primer sequences (cDNA_SSP_VNP.fas). The resulting 8.7 M reads were combined into a single archive and later used for gene annotation (see below). We also produced a non-redundant set of sequences to represent potential genes in the krill genome. Here we extracted long and high-quality cDNA reads

(length > 500 bp and Q > 12) from both runs and clustered them into putative genes with isONclust2[87] using the "-x sahlin" mode. We then error-corrected the clustered cDNA reads with isONcorrect v0.0.6[88]. Lastly, for each error-corrected cluster with four or more cDNA reads we generated a consensus sequence with vsearch v2.14.1[89]:

vsearch --cluster_fast <clusters.fa > --threads 40 --id 0.95 --cluster-out_id --clusterout_sort --sizeorder --sizeout --consout <clusters.fa.cons>

This produced 24,632 consensus sequences with $N_{50} = 1{,}810$ bp. These non-redundant transcript sequences were later used for RNA-based scaffolding (see below).

**Short-read RNA-seq processing and transcriptome assembly.** The Illumina RNA-seq data from the reference specimen and two published RNA-seq archives, TI787-MN-M (adult male; SRR3657321) and TI787-MN-F (adult female; SRR3657320), were trimmed for adapters and base qualities using Trim Galore! v0.6.1 (https://github.com/FelixKrueger/TrimGalore/) and Cutadapt v2.3[90]:

trim_galore -j 8 --gzip --length 50 --trim-n --paired

All read pairs and 89–99% of bases were retained. The trimmed reads were later used in genome assembly scaffolding and annotation (see below). We used Trinity v2.5.1[80] to assemble a transcriptome from all 459 M trimmed read-pairs from the reference specimen, producing 573,869 with $N_{50} = 900$ bp. Using BUSCO v3.0.2b[91] with the Arthropoda odb9 lineage set, we detected almost all expected genes (C:97.2% [S:42.4%,D:54.8%], F:1.6%, M:1.2%, n:1066). A subset of 500 bp or longer transcripts were used in scaffolding (see below). We processed all transcripts with Trinotate v3.1.1[92] and identified 60,677 possibly coding transcripts with best hits against metazoan templates from parsing the "Trinotate.xls" output and used these transcripts for gene annotation (see below). A subset of 16,509 non-redundant transcripts with best hits against arthropods were used to measure assembly quality scores (see below).

### Mitochondrial re-assembly

We re-assembled the mitochondrial chromosome for the K20 reference specimen similarly as described above. PromethION reads identified during the screening process and that were also between 1–16 kb long were used to assemble the chromosome in Canu v2.0-development[81] with "genomeSize=17k" that produced a 26.1 kb contig that the assembler suggested was circular. We next mapped the 369,946 mitochondrial long-reads to the sequence with minimap2 v2.17-r941[82] and polished it with Racon using the "variants" and "vcf2fasta" subcommands. We circularized the sequence manually, producing a mitochondrial sequence that was 17,840 bp. We then mapped a subset of the 10x Chromium data (656,771 read pairs) to the chromosome with BWA and performed one round of polishing with Pilon v1.23 with the "--fix all" setting, followed by a round of polishing with K20 RNA-seq data. The final mitochondrial sequence was 17,944 bp with %GC = 28. Automated annotation was again carried out with the online MITOS2 platform as before and features were visualized with DNAPlotter[93] (Supplementary Fig. 3).

### Genome assembly

The hybrid genome assembly underwent multiple steps of polishing, scaffolding and refinements, in which we prioritized achieving high gene completeness and gene contiguity. We mainly used genetic information from the reference specimen, but also included data from other specimens for specific tasks (such as identifying redundant haplotigs). Nanopore long-reads were used for assembly, polishing (reducing base-level errors) and scaffolding, while 10x barcoded linked-reads and RNA data were used for polishing (in highly expressed regions) and scaffolding.

**Genome assembly.** Scaffolding was performed with top priority on recovering gene bodies using RNA from the reference specimen and additional RNA-libraries[94], followed by genome-wide scaffolding using Nanopore synthetic mate-pairs with FAST-SG[95] and 10x Chromium barcodes with Scaff10x. Some inadvertently removed contigs with genes were manually re-inserted into the genome. Screens against the SILVA ribosomal database[96] were used to isolate contigs from bacterial contaminants. The assembly was carried out in 13 steps and we used BUSCO[97] and genecovr (https://github.com/NBISweden/genecovr) throughout the pipeline to assess gene detection, completeness and sequence error.

**Assessment of genome assembly quality and completeness.** We used BUSCO v3.0.2b with the Arthropoda odb9 lineage set or BUSCO v5.0.0[97] with the Arthropoda odb10 lineage set, respectively, to track improvements in gene detection rates between genome assembly versions, and to assess the overall completeness and levels of duplication in the final genome. In addition, we assessed overall gene completeness by mapping a set of 16,509 trinity transcripts to the assemblies using gmap version 2020-06-30[98]. We parsed the psl output and summarized various statistics related to mapping quality, including number of insertions, number of mismatches, number of distinct contig hits, alignment length and alignment depth, to obtain "gene coverage" statistics for each transcript. Compilation of the statistical summaries and plots were implemented in a custom R package genecovr:

- https://github.com/NBISweden/genecovr.

**Assembly v1: Production of a preliminary assembly with the wtdbg2 assemble.** To assemble the krill genome, we used the wtdbg2 v2.5 assembler[99] and 6–300 kbp long reads with Q-scores ≥ 10 selected with NanoFilt v2.6.0[100] and that had been pre-screened for internal adapters or mitochondrial sequence, corresponding to ~16× depth of coverage across the genome. We used a short kmer setting ("-p 17"; as recommended for noisy data by the developers) and sparse kmer subsampling ("-S 2") for sensitive read analysis and binning. Assembling the genome using these settings ("-A -p 17 -S 2 -s 0.05 -L 6000 -g 18 g -t 168") took 16 days on a 168 core machine with 4.7 TB RAM. Consensus contigs were exported with wtdbg-cns, resulting in genome assembly v1, which spanned 21.8 Gbp across 797,490 contigs with $N_{50} = 41$ kbp.

**Assembly v2: Long-read polishing with Racon.** The PromethION long-reads were mapped in parallel chunks to the v1 genome assembly using minimap2 v2.17-r941 set to use a split prefix and without secondary alignments ("-L -ax map-ont -y -t 16 --secondary=no --split-prefix"). Data was first saved as gzipped SAM files and then sorted according to coordinates into BAM files with samtools v1.10[101], followed by merging into a single BAM file with samtools. The genome assembly was split into 248 100 Mbp FASTA chunks with a custom Perl script and the mapped reads were partitioned into corresponding SAM and FASTQ files using samtools. We used one round of Racon v1.4.11 with the average base quality threshold set to 8 ("-q 8") to polish all 248 chunks in parallel. The polished contigs were concatenated into genome assembly v2.

**Assembly v3: Long-read polishing with Medaka.** We followed the guidelines from the medaka source documentation (https://github.com/nanoporetech/medaka#improving-parallelism) that suggests running the medaka components independently and in parallel for large genomes. Briefly, the PromethION long-reads were mapped in 250 parallel chunks to the v2 genome assembly using minimap2 v2.17-r941 with parameters "-x map-ont --MD --secondary=no -t1". This was followed by the consensus algorithm medaka consensus with parameters "--model r941_flip235 --batch_size 100 --threads 1" to generate models over each assembly chunk. Finally, consensus sequences were generated with medaka stitch and merged to obtain assembly v3.

**Assembly v4: Short-read polishing with Pilon.** The 10x Chromium paired-end linked-reads were contained in 16 paired FASTQ libraries that were mapped to the v3 genome assembly with BWA v0.7.17[102,103]. First, the genome reference sequence was indexed ("bwa index -a bwtsw") and the data was then mapped with the "bwa mem" algorithm, piping the output to samtools which sorted the data on the fly before saving a BAM to disk. The 16 BAMs were merged with samtools. As in the Racon step above, both the reference genome and the BAM were split according to chunks of contigs spanning 100 Mbp. Polishing was performed in parallel with Pilon v1.23 set to polish putative base-level errors and specifying a diploid dataset ("--fix snps,indels --diploid"). The short-read polished contigs were concatenated into genome assembly v4.

**Assembly v5: Purging haplotigs with Purge_haplotigs.** Since our genome assembly was about 3 Gbp larger than the expected genome size (21.8 Gbp vs ~19 Gbp), we next sought to identify and remove potential haplotigs from the assembly using Purge_haplotigs v1.1.0[104]. The tool uses sequence similarity among contigs as well as a bimodal signature of read mapping depths to isolate and remove alternative alleles of the same locus from highly heterozygous diploid genome assemblies. It also identifies "junk" contigs with extreme mapping depths (low or high) that may be assembly artifacts or otherwise intractable in downstream analyses.

We first mapped the ~32× PromethION long-reads using Minimap2 to the v4 assembly, as in previous steps, and then used the "purge_haplotigs hist" command to generate a depth-of-coverage histogram across the genome. However, this relatively low-coverage dataset did not result in a clearly bimodal mapping profile with 1x and 0.5x mapping peaks on its own. We therefore also mapped the 10x Chromium linked-reads and a preliminary set of ~140× population resequencing data to the genome using BWA, in order to use all available data to specify depth thresholds to flag offending contigs. The short-read BAMs were marked for duplicates with Picard MarkDuplicates. We then exported per-base depth-of-coverage data running "samtools depth" across all positions in the genome assembly ("-a") and generated a depth histogram with a discernable "1x" peak at ~150×, presumably corresponding to properly haplotype-fused contigs. The expected position of the 0.5x peak at half of this depth was less clear and possibly intermixed with genomic regions of low mappability. Given the depth profile, we then configured Purge_haplotigs to use 20× and 240× as lower and upper cut-offs, respectively, and 125× as the mid-point. We specified that contigs with 100% or less of their depths at diploid level of coverage to be flagged as suspected haplotigs (i.e. to comprehensively evaluate all contigs) and that those that had depths outside of the upper/lower cut-offs across > 50% of their lengths to be flagged as junk:

purge_haplotigs cov -l 20 -m 125 -h 240 -s 100 -j 50 -i <coverage_file> -l 20 -m 125 -h 240 -o <coverage_stats_output>

Purge_haplotigs infers contig similarities from the proportion of sequence that can be aligned between two contigs using minimap2, and can be set to disregard alignments across repeat sequence. We therefore produced a preliminary catalog of repeat sequence intervals in BED format using Red[105], masking 11.7 Gb (54%) of the genome. We used the default alignment score cut-off ( ≥ 70%) to mark contigs for reassignment as haplotigs:

purge_haplotigs purge -align_cov 70 -repeats <repeats_file>

The purge subcommand executes the script "purge.pl". We modified this script to increase performance in the handling of our large, repeated and relatively fragmented genome assembly by: implementing faster parsing of tabular data; a string-based instead of array based data structure for hit summaries; and faster internal repeat-handling instead of executing the external tool bedtools from within the script. This code is available at: https://github.com/andreaswallberg/Ecological-Genomics-Northern-Krill.

Using these settings, we purged 168,316 contigs ($N_{50}$ = 13,074 bp) spanning 1.95 Gbp as potential haplotigs and 49,996 contigs ($N_{50}$ = 14,236 bp) spanning 565 Mbp as junk/artifact contigs, respectively, removing 12% of the original genome assembly. The remaining main assembly had 586,516 contigs ($N_{50}$ = 46,208 bp) spanning 19.18 Gbp and was taken as genome assembly v5.

**Assembly v6: Mixed-read polishing with HyPo.** We aimed to polish the genome further after purging haplotigs and mapped both the PromethION long-reads and 10x Chromium linked-reads from the reference specimen back to the genome as above. We next split the genome assembly, BAMs and FASTQ reads (using samtools) into ten chunks of contigs spanning 2 Gbp of the genome each and ran the polisher HyPo v1.0.3[106] separately on each chunk, which uses both short and long reads to polish contigs in a single run. The polished contigs were taken as genome assembly v6.

**Assembly v7: Scaffolding the contigs with Trinity transcripts using L_RNA_Scaffolder.** At this stage, quality assessments indicated that we had many broken gene models with exons distributed across more than one contig. Preliminary trials suggested that scaffolding contigs together based on information in RNA-mappings before long-read or linked-read DNA scaffolding recovered more complete gene models for this genome than vice-versa. We started by scaffolding contigs with non-redundant full-length cDNA transcripts ($n$ = 24,632; see section above) and Trinity-assembled transcripts, i.e. contiguous RNA evidence that was assembled independently from the genome assembly itself, using L_RNA_Scaffolder[107]. We first filtered the Trinity transcripts to only include the longest isoform of each gene and kept only the longest isoforms that were > 500 bp long ($n$ = 73,422; $N_{50}$ = 1,482 bp). L_RNA_Scaffolder typically uses BLAT to infer query alignments, which does not support genomes larger than 4 Gbp. We therefore instead used NCBI BLAST v2.2.31 +[108] to build a database:

makeblastdb -dbtype nucl -in <reference>

We then queried both sets of transcripts using MEGABLAST:

blastn -query <transcripts.fa > -task megablast -db <reference > -outfmt 5 -perc_identity 80 -max_target_seqs 100 -num_threads 36

We then converted the XML output into the expected PSL format using UCSC blastXmlToPsl v412 (http://hgdownload.soe.ucsc.edu/admin/exe/). L_RNA_Scaffolder joined contigs with shared transcripts to generate 12,277 scaffolds. The scaffolds had $N_{50}$ = 158,477 bp, compared to $N_{50}$ = 44,758 bp among the unscaffolded contigs ($n$ = 553,073). Scaffolds and contigs were joined into genome assembly v7.

**Assembly v8: Scaffolding with short-read RNA-seq data with a first pass of BESST_RNA.** We next used BESST_RNA (https://github.com/ksahlin/BESST_RNA) to perform additional RNA-based scaffolding using paired-end RNA-seq data. We first mapped the 755 M processed Illumina short-read RNA-seq read-pairs derived from the reference specimen itself and two published specimens (one male with ID TI787-MN-M; one female with ID TI787-MN-M; Supplementary Table 1; Supplementary Fig. 2)[94] against the assembly with HISAT2 v2.2.1[109] (mapping rates 81–86%) and merge the data into a single BAM file using samtools. We then performed scaffolding with BESST_RNA requiring three or more witness links with a maximum of 200,000 bp distance ("-e 3 -T 200000 -k 500 -d 1 -z 10000 -g 1"), which resulted in a total of 15,519 RNA-based scaffolds (including those from the previous step) and reduced the number of sequences to 557,087 in genome assembly v8.

**Assembly v9: Scaffolding with long-read DNA data using FAST-SG +ScaffMatch.** We hypothesized that there could be residual unresolved long-range contact information in the PromethION DNA data and used the Fast-SG scaffolder[95] with KMC v3.0.0[110] for k-mer

counting to accomplish further scaffolding. First, we extracted a subset of long-reads ≥ 5 kbp and with average Q-scores above 10 from the read pool. Next, we generated synthetic mate-pairs in Fast-SG with insert lengths ranging from 4,000 to 40,000 bp (n = 10) using 32 bp kmer matches spaced along reads and the genome to enable downstream detection links between contigs. The mate-pair settings were specified in a single-line read configuration file:

long ont <reads.fq> 4000,5000,6000,7000,8000,10000,15000, 20000,30000,40000 1

Fast-SG was run as:

FAST-SG.pl -k 32 -l <read_configuration.txt> -r <genome_assembly.fa> -p results -c $KMC -t 40

This produced two SAM files per synthetic mate-pair group, containing perfect forward/reversed kmer matches along reads and contigs at the approximate insert lengths specified for each insert group. The empirical mean and standard deviation of the insert lengths was computed from the first 1 M observations in each group as per the FAST-SG manual (https://github.com/adigenova/fast-sg/wiki/Hybrid-scaffolding-of-NA12878).

The statistics and SAM mappings were next processed in Scaff-Match v0.9[111] to generate and resolve a scaffolding graph in order to scaffold the genome assembly:

python2.7 $PATH/scaffmatch.py -w ONT-K32 -c <genome_assembly.fa> \

-s 1461,1734,1907,2137,2194,2646,3736,7711,10503,13425 \ # std-dev
-i 3957,4964,5965,6958,7938,9921,14634,16513,26175,34772\# mean
-p fr,fr,fr,fr,fr,fr,fr,fr,fr,fr \
-m1 <ont.I4000.fwd.sam> , <ont.I5000.fwd.sam> , … \
-m2 <ont.I4000.rev.sam> , <ont.I5000.rev.sam> , … \
2 > &1 | tee scaffmatch.log

Fast-SG+ScaffMatch produced a scaffolded genome assembly spanning 247,751 sequences ($N_{50}$ = 142,666 bp), out of which 131,888 were scaffolds and 115,863 were contigs, and was taken as genome assembly v9.

**Assembly v10: Scaffolding with linked-read DNA data using Scaff10x.** Having produced scaffolds with RNA data and long-reads and reduced the overall number of sequences in the genome assembly, we next explored the possibility to scaffold sequences based on signatures of shared 10x Chromium linked-read barcodes between the outer edge regions of scaffolds and contigs. Scaffolding was performed using Scaff10X v4.2 (https://github.com/wtsi-hpag/Scaff10X).

First, we mapped the linked-reads to the genome assembly using BWA, as in previous steps. For performance and memory reasons, we next pre-filtered the BAM file to only include reads mapping to the outer 20 kbp regions of scaffolds or contigs using samtools and a custom Perl script. To reduce the influence of ambiguous mappings to highly repetitive sequences in the outer regions, we also filtered the reads to only include read-pair mappings with MAPQ ≥ 20 in the same script. We then ran one iteration of Scaff10x, and configured the program to require at least six reads per barcode ("-reads 6"), eight shared barcodes as evidence of links between sequences ("-link 8"), edge length to match our estimated linked-read lengths ("-edge 20000"), with other settings at their default values:

scaff10x -nodes 20 -gap 100 -longread 1 -reads 6 -link 8 -edge 20000 -plot <mappings.bam>.png -bam <mappings.bam> <genome_assembly.fa>

This resulted in scaffolded genome assembly v10, spanning 219,207 sequences with $N_{50}$ = 214,496 bp, including 108,310 scaffolds.

**Assembly v11: Scaffolding with short-read RNA-seq data with a second pass of BESST_RNA.** After scaffolding the genome assembly with long-read and linked-read DNA data, we applied a second round of RNA scaffolding using BESST_RNA and RNA-seq data, as above, to find gene-based links that were not resolved in previous steps.

BESST_RNA reported that 2,294 new scaffolds had been formed. This round of scaffolding further reduced the number of sequences from 219,207 in v10 to 216,722 ($N_{50}$ = 220,983 bp) in genome assembly v11, out of which 106,444 sequences were scaffolds.

**Assembly v12: Short-read polishing with Pilon using high-coverage RNA-seq data.** The Illumina RNA-seq data from the reference specimen was re-mapped to the genome assembly (88.3% mapping rate) and one final round of base-level polishing Pilon ("--fix snps,indels") was applied across transcribed regions with at least 100× depth of coverage ("--mindepth 100"), resulting in sequences with minor adjustments that were taken as genome assembly v12.

**Assembly v13: Finishing the assembly by removing contaminants and re-inserting contigs.** Some qualitative adjustments were carried out to reassign sequences between the main genome assembly and other classes of sequence in order to maximize gene completeness and remove putative non-krill contaminations and mitochondrial artifacts. We screened the main assembly (v12), as well as the "artifact" (~565 Mb) and "haploid" (~1.95 Gb) sequences classified by purge haplotigs (produced in genome assembly v5) to identify candidate sequence for removal or inclusion.

**Reintroduction of krill genes in to the main assembly.** We identified missing genes in the main assembly using both comparative and experimental evidence. By running BUSCO on the artifacts and haplotigs we identified two complete single-copy BUSCO genes that had no corresponding BUSCO tblastn hit in the main assembly. In addition, we used GMAP to identify ten contigs from the haplotig class that had unique Trinity transcripts mapped with high quality (>90% identity across >90% of the transcript) but with no hits in the main assembly. We considered these 12 contigs to have been inadvertently misclassified and transferred them back into the main assembly.

Ribosomal gene sequences occur in high copy numbers in many metazoans and may be difficult to target accurately with short-read data. We therefore used the nucleotide-nucleotide BLAST (blastn) and the extensive SILVA LSU and SSU ribosomal RNA databases (release 132)[96,112] to identify artifact sequences with ribosomal loci from the krill. In the majority of contigs where hits were detected, the top scoring krill alignment was against the expected *M. norvegica* templates (LSU accession AY744900.1.2423 [2,423 bp]; SSU accessions AY781434.1.1853 [1,853 bp] or DQ900731.1.1884 [1,884 bp], respectively), although sometimes templates for other krill had better matches (typically from the Antarctic krill *Euphausia superba*), which occurred in particular when matches were short and fragmented. For example, without considering the quality and lengths of the reported hits against the main assembly and the artifacts, the *M. norvegica* LSU template had the best krill match in n = 70/96 [73%] of cases, while the SSU templates were the best krill matches in 15/27 [56%] of cases. Requiring hits to be at least 1,500 bp with 98% identity against the template raised the on-target *M. norvegica* best-hits to 91% (n = 10/11) and 100% (n = 8/8) for LSU and SSU. We found that most sequences with long motifs matching *M. norvegica* ribosomal templates (>1,500 bp; >90% identity) had been placed into the artifact class (LSU: $n_{artifacts}$ = 14 vs $n_{main}$ = 5; SSU: $n_{artifacts}$ = 10 vs $n_{main}$ = 5), likely due to aberrant mapping depths across these multi-copy loci. We transferred these artifact contigs (n = 18) with long matches back into the main assembly.

**Detection of contaminants.** We searched the SILVA LSU and SSU ribosomal RNA databases to identify possible eukaryotic or bacterial contaminants using blastn and considering the top-scoring alignment for each assembly sequence. Among contigs flagged as artifacts, we detected long and high-identity ribosomal matches (>1,500 bp and >90% identity) against *Enterobacter ludwigii/cloacae, Lactobacillus*

*reuteri* and *Delftia tsuruhatensis*. Querying the Nanopore sequence data itself against the SILVA database (see below), we re-classified the *Delftia* matches as *Delftia acidovorans*. We also identified several sequences in the main assembly with partial SSU low-identity hits against *Aphelidium desmodesmi* (n = 204; average identity=81%), a eukaryotic endoparasite of freshwater green algae. However, this SSU accession (KY249641.1.3521 [3,521 bp]) also contains a partial LSU sequence with relatively high identity towards euphausiids (~85% identity across 550–650 bp). A combined search using both SSU and LSU simultaneously completely masked the hits against this accession in favor of *M. norvegica* LSU hits.

We downloaded the corresponding bacterial genome sequences from NCBI Genome Database (CP027618.1 for *Enterobacter cloacae* [5.0 Mbp genome]; CP015408.2 *Lactobacillus reuteri* [2.0 Mbp] and CP019171.1 for *Delftia acidovorans* [6.6 Mbp], respectively). We then used blastn to screen the krill genome assembly for contigs and scaffolds matching these sequences. Among artifact contigs, we recovered multiple high-scoring alignments between assembly sequences and the *Enterobacter* (n = 68; average alignment length=48 kb; average identity=92.5%) and *Delftia* (n = 75; average alignment length=44 kb; average identity=92.4%) genomes, respectively. Among the sequences in the main assembly, we detected less *Enterobacter* (n = 7; average alignment length=24 kb; average identity=99.2%; genome coverage=3%) and *Delftia* (n = 12; average alignment length=88 bp; average identity 96.2%; genome coverage=4%). Matches against the *Lactobacillus* tended to score lower, covering only 0.6% of the bacterial genome among artifact sequences (n = 7; average alignment length=1.9 kbp; average identity 86%) and 2% among main assembly sequences (n = 17; average alignment length=2.1 kbp; average identity 97%).

We then queried up to 1 M Nanopore reads produced by each PromethION flow cell and sequencing-run (original or flushed) against each bacterial genome (in total, 14.8 M reads out of 84.4 M reads) using blastn and estimated that we had sequenced about 73 k reads from *Enterobacter* for a total of about 60 Mb or 12× average coverage across its genome, 66 k reads of *Lactobacillus* (~82 Mb; 40× coverage) and 14 k reads of *Delftia* (~180 Mbp; 27× coverage). Taken together (assuming no overlap between the datasets) we estimate that about 0.02% of reads and 0.05% of data was derived from these bacteria rather than the krill, respectively. Because of the low frequencies and sporadic appearance of reads among flow cells, we hypothesize that the bacteria were technical contaminants rather than being naturally associated with the krill itself. *Delftia acidovorans* has previously been associated with biofilm contamination of BluePippin size selection cassettes[113]. We used a basic blastn criterion to move main and artefact assembly sequences with at least one top-scoring alignment with >90% identity towards any of the bacteria across 500 bp into a "contamination" class (*Enterobacter* n = 51; *Lactobacillus* n = 9; *Delftia* n = 54), spanning about 98–99% of aligned sequence.

**Flagging residual mitochondrial sequence.** We had already screened the Nanopore dataset for mitochondrial reads against a draft assembly of the mitochondrial chromosome. Nevertheless, we probed the genome assembly for sequences matching the *M. norvegica* mitochondrion with >90% identity across at least 2 kb (longer than any individual mitochondrial gene) and detected 15 sequences harboring either mitochondrial genes or fragments of the AT-rich control region. Three of these were scaffolds, each of which with a single contig containing the long match but with short matches on one or more other contigs. We split these scaffolds and extracted only the offending contig while transferring the remaining contigs back into the assembly. We can not exclude that some of these sequences are nuclear mitochondrial DNA (NUMTs) rather than misassembled sequences and therefore put these sequences into their own mitochondrial artifact class.

## Genome annotation

We first applied multiple tools to find simple and interspersed repeats, using structural signatures or protein domain homology for classification of transposable elements, and then produced a non-redundant repeat library to characterize the repeat landscape. To annotate genes, we generated gene models using RNA-seq data and cDNA or Trinity transcripts. We also mapped transcripts or gene sequences from seven other Malacostracans to find unexpressed genes and consolidated overlapping gene models into common loci, from which high-scoring non-redundant isoforms were selected as the canonical representatives of genes.

## Repeat annotation

We aimed to build a custom non-redundant repeat library for the Northern krill and use it to estimate the distribution of repeats across the genome. We applied several specialized or general standard repeat detection pipelines to annotate simple, tandem and interspersed repeats and transposable elements (TEs) in the krill genome, using structural or homology-based searches against characteristic motifs, de novo detection and de novo assembly of repeats from short-reads. Intermediate repeat libraries were interrogated against the genome and/ or for TE domains and masked or clustered to remove redundancy, before a final non-redundant interspersed repeat library was incrementally produced and used to annotate the genome. Some intermediate steps to filter or rename repeats based on the output from the respective tools were facilitated with custom Perl scripts. The procedure to detect interspersed repeats was inspired by two online protocols:

1. MAKER WIKI: http://weatherby.genetics.utah.edu/MAKER/wiki/index.php/Repeat_Library_Construction-Advanced
2. Berriman Lab Group (Sanger Institute) protocol on Protocol Exchange[114]: https://doi.org/10.1038/protex.2018.054

**Detection of simple repeats and low complexity regions.** We used SciRoKo v3.4[115] with default settings to estimate the degree of microsatellites in the genome assembly and identify common microsatellite motifs. We then used SDUST v0.1 (https://github.com/lh3/sdust), a fast reimplementation of the symmetric DUST algorithm in NCBI Dustmasker[116], using defaults to estimate the proportion of simple repeats and low-complexity regions in the genome. Lastly, we used TideHunter v1.4.4[117] to detect tandem repeats with a minimum period of 6 bases that occurred in at least two copies ("-p 6 -c 2") and estimate the overall tandem repeat content in the genome assembly.

**Detection and characterization of LTRs using structural searches with LTR_Finder and LTRharvest.** Characteristic structural features of LTR (Long Terminal Repeat) retrotransposons were searched with LTR_Finder v1.07[118] using the LTR_FINDER_parallel wrapper v1.1[119] and LTRharvest[120]. The output of each tool was processed with LTR_retriever v2.9.0[121] to remove putative low-quality false positive LTR hits. Internally, LTR_retriever used RepeatMasker v4.1.2-p1[122] and CD-HIT v4.8.1[123] ("-cdhit [-c 0.80 -n 5 -M 0 -aS 0.80 -G 0 -g 1]"). We used a mutation rate inferred from snapping shrimp in[30] of 2.64e⁻⁹ substitutions per site per year ("-u 2.64e-9"), "-missmax 10000" to allow LTRs to span contig gaps, "-notrunc" to discard truncated LTRs and nested LTRs and "-noanno" to skip whole-genome LTRs annotation at this stage.

LTR_retriever filtering resulted in intermediate libraries with 10,740 LTRs from LTR_Finder (headers tagged "FIN") and 4,235 LTRs from LTR_Harvest (headers tagged "HAR"), respectively. We next mapped these motifs back to the genome using MEGABLAST in RepeatMasker RMBLAST v2.11.0 and carried forward only those motifs that had two or more hits with >80% identity towards the genome across >80% of the repeat motif ("80/80" rule)[124]. Many LTRs are autonomous and contain several genes to facilitate their retrotransposition. The LTRs in each library were therefore characterized

through homology to known LTRs or LTR protein domains using multiple tools:

1. RepeatClassifier in the RepeatModeler v2.0.2 suite[125] to find matches against sequences in the RepBase RepeatMasker Edition or Dfam DNA databases (version Dfam_3.5), respectively, using the RMBLAST v2.11.0 search engine and default search thresholds.
2. TEsorter v1.3[126] to detect matches against TE protein domains using the REXdb-metazoa database (v3)[127].
3. TransposonPSI v08222010 (http://transposonpsi.sourceforge.net/) with the BLAST v2.2.26 search engine using the "PSI-BLAST" task[128,129] to query the LTRs against TE family profiles using the default search parameters.
4. HMMER3 v3.3 (http://hmmer.org/) searches against Dfam LTR profiles. Here we first generated a list of profiles matching LTRs from the curated release Dfam_3.5 using the command:
   grep -P "(\tLong\sterminal\srepeat|LTR|Gypsy|Copia|retro-transposon)" Dfam_curatedonly.hmm.list.csv | grep -v -P "(L\d + | LINE)" > Dfam_curatedonly.hmm.list.csv.LTR.csv
   We then used hmmfetch to extract the actual profiles and hmmsearch ("--notextw --cpu 40 -E 1e-5 --domE 1e-5") to search a version of the LTRs where we had first masked out simple repeats with dust to reduce spurious hits driven by micro-satellite motifs.

We next used a custom Perl script to parse the output from the four tools and only keep LTRs with a significant hit detected using at least one of them. The LTRs were classified according to the detected LTR type (superfamily or similar level) and tagged with the detection-tool in the following order of priority: i) RepeatClassifier ("ReC"); ii) TEsorter ("TEs"); iii) TransposonPSI ("PSI"); and iv) HMMER+Dfam ("DFA").

LTR retrotransposons have two similar 5′ and 3′ LTR regions and we divided the LTRs into a pool of repeats with highly identical LTR regions (99–100% identities; putatively "young" motifs) and a pool of repeats with more divergent regions (85–99% identities; possibly older repeats). Sequences in each pool were then translated into peptides by LTR_retriever's Six-frame_translate.pl tool and then searched separately with hmmsearch (--notextw --cpu 40 -E 0.01 --domE 0.01) and the GyDB2 database[130] for complete Copia or Gypsy LTRs with all expected internal genes still present, i.e. those with simultaneous hits against all five typical protein domains: Capsid "GAG", Aspartic proteinase "AP", Integrase "INT", Reverse transcriptase "RT" and RNase H "RH". We then applied a step-wise procedure to reduce redundancy in the LTR libraries, while putting top priority on complete LTRs with high LTR region identities:

1. We combined the complete LTRs with 99% LTR region identities from LTR_Finder and LTRharvest into a single set and removed redundant 80/80-hits with CD-hit (-c 0.80 -n 5 -M 0 -aS 0.80 -G 0 -g 1 -T 40).
2. We then combined the complete LTRs with 85–99% identical LTR regions from LTR_Finder and LTRharvest and masked this set with RepeatMasker using the preceding non-redundant set from step 1. LTRs with <80% sequence masked were kept and clustered to remove redundancy with CD-hit as above. The representative sequences were then concatenated to the set from step one to create a non-redundant set of "complete" LTRs.
3. We next combined the incomplete LTRs with 99% LTR region identities and repeat-masked them with the "complete" LTR library, adding only <80% masked and CD-hit clustered non-redundant LTRs to the library. This was followed by a final iteration of masking and adding LTRs using the incomplete 85–99% LTR library as above.

Our final non-redundant and homology-based LTR library contained 365 full-length or near full length retrotransposon sequences (length N50 = 6,421 bp) (Supplementary Data 3).

**Detection of diverse transposon domains using TransposonPSI.** We carried out PSI-BLAST homology searches using the broad library of transposon ORF profiles (including both LTRs, LINEs, DNA transposons and other elements) in TransposonPSI v08222010 (using BLAST v2.2.26) across the whole krill genome assembly. For efficiency, the genome assembly was first split in 100 chunks, and the chunks were scanned in parallel in TransposonPSI. High divergence between query and profile may result in fragmented HSP (High Scoring Pairs) hits interspersed by non-matching sequence. We compiled a preliminary repeat library from the "*.TPSI.allHits.chains" files, which contains chains of one or more collinear HSPs hits against transposon profiles. i.e. series of HSPs that align collinearly between a specific TE ORF and the genome, that are not interrupted by ORFs of other TEs. As in the LTR scans above, we then queried the library against the genome and kept only repeats with two or more 80/80-hits against the genome (n = 190,408). Based on the initial classification assigned by TransposonPSI, we subdivided the repeats into broad groups: LTRs, LINEs, DNA transposons and Rolling-Circle transposons (Helitrons). LTRs were masked with the preliminary LTR library (see above) using RepeatMasker and only sequences with <80% masked positions were kept. The sequences of each group were then translated into peptides using Six-frame_translate.pl and either the DNA or peptide sequence was then used to re-classify the repeats using the programs and databases also used to classify the LTRs in the previous section:

1. hmmsearch was run with both the GyDB2 database to detect Copia or Gypsy domains and the REXdb-metazoa database to detect other TE domains (--notextw --cpu 40 -E 0.01 --domE 0.01). Putative LINEs, DNA transposons and helitrons were kept only if they did not have significant matches against LTR domains. The sets were concatenated and clustered with CD-HIT to remove redundant sequences (-c 0.80 -n 5 -M 0 -aS 0.80 -G 0 -g 1 -d 0 -T 40) before proceeding to the next step.
2. RepeatClassifier was run using both RepBase RepeatMasker Edition and Dfam DNA databases, as above.
3. TEsorter was run (-st nucl -p 40 -tmp tmp -eval 0.01) with the GyDB2, REXdb-metazoa and REXdb-tir dabases.

We used a custom Perl script to parse the output from the classification tools. Putative LINEs, DNA transposons and helitrons were kept only if they had at least one match against the expected group (class/order) and did not have significant matches against LTR domains. The repeats were classified according to the detected type (i.e. superfamily) and tagged with the detection-tool in the following order of priority: i) RepeatClassifier ("ReC"); ii) TEsorter ("TEs"); iii) TransposonPSI ("PSI", the original classification of the repeat). Compared to the refined structural LTR library, we detected more but shorter LTR fragments using this approach (Supplementary Data 3).

**Detection of transposable elements through de novo repeat assembly with dnaPipeTE.** Repeats and TEs that occur at high frequency in the genome can be detected through de novo assembly of low-coverage short-reads. This approach is implemented in dnaPipeTE[131], which uses Trinity to assemble repeats from short-reads independently of a reference genome and BLASTN to classify them against its own database. We extracted 16 independent batches of 10 million single short-reads from our 10x Chromium Illumina read libraries and assembled repeats independently for each batch using dnaPipeTE v1.3.1 (-genome_size 19000000000 -genome_coverage 0.079 -sample_number 2). We then parsed the "one_RM_hit_per_Trinity_contigs" output-file from each run with a custom Perl script and kept only those Trinity-assembled repeat contigs where a reported BLASTN match against a dnaPipeTE repeat template spanned >20% of length of both the contig and the template. The repeats of each run were concatenated into a single preliminary library (n = 4,799) and

subdivided into major groups (DNA transposons, LTRs, LINEs and SINEs). As in the preceding repeat scans, we queried each group against the genome and kept only repeats with two or more 80/80-hits against the genome (n = 3,381). From manual inspection, we noticed that candidate TEs frequently consisted nearly exclusively of simple repeats / microsatellites. We therefore subjected also this preliminary library to reclassification with the same tools used to process the previous repeats:

1. The sequences of each group (DNA transposons, LTRs, LINEs and SINEs) were masked with RepeatMasker using the two LTR and TransposonPSI libraries from previous annotation steps, respectively, and only sequences with <80% bases masked were kept. Each group was clustered separately with CD-hit as above to reduce redundancy.
2. Each group was re-classified using the same tools and procedures outlined in the TransposonPSI pipeline above.

Our filtering and clustering resulted in 311 dnaPipeTE repeats being added to the growing repeat library, most of which appeared to be relatively short TE fragments (Supplementary Data 3).

**Detection of transposable elements from high-frequency motifs using RepeatModeler2.** We ran RepeatModeler2[125] across a random subset of contigs corresponding to 10% of the krill genome assembly and detected n = 3,799 putative consensus repeat sequences occurring at high copy numbers. We applied a similar approach as in preceding steps to remove redundant repeats from this preliminary library:

1. The sequences were masked with RepeatMasker using the three finalized LTR, TransposonPSI and dnaPipeTE libraries, respectively, and kept only RepeatModeler sequences with <50% bases masked. These were then clustered with CD-hit as above to reduce redundancy, generating n = 2,091 repeat sequences.
2. The sequences were re-classified using RepeatClassifier as above. Only 156 could be annotated with RepeatClassifier, suggesting that RepeatModeler2 tended to detect high-frequency repeats with low homology to known transposable elements motifs in the repeat databases, possibly due to excess uncorrected sequence error or evolutionary degeneration or divergence among repeat copies detected throughout the genome.
3. To improve the annotation rate of this library, we scanned individual repeat copies detected by each consensus sequence. First, we mapped consensus sequences to the genome assembly with MEGABLAST and kept up to n = 1,000 hits against the genome with >90% identity across 80% of the consensus sequence. We then queried each copy against a curated protein database in RepeatMasker using its utility RepeatProteinMask (which uses BLASTX), registering alignment scores with p-values < 0.01. We summed the alignment scores across copies and annotated the consensus sequence with the order/superfamily classification of the RepeatProteinMask template with highest total score (i.e. the majority vote). We required that the majority vote was based on at least 10 observations and otherwise left consensus sequences unannotated. This approach added annotations to 644 consensus repeats, more than double the annotation rate compared to annotating the consensus sequences directly with RepeatProteinMask (n = 254).

Our RepeatModeler library thus spanned 2,091 interspersed repeat sequences, out of which 800 were annotated to class and subfamily (Supplementary Data 3).

**Evaluation and validation of the repeat library.** We concatenated all libraries with interspersed repeats into a single library (n = 10,908; 13.8 Mb). As a means to evaluate the annotation and characteristics of the library with independent information, we queried the library

against a set of 1,000 common transposon-associated protein domains derived from the NCBI Conserved Domain Database (CDD). The domain data was available in the TransposonProteinNCBICDD1000 tool in the TransposonUltimate annotation suite[132]. We used the "Reverse Position-Specific BLAST" RPSBLASTN tool to query the library and detect significant hits against pre-calculated Position-Specific Score Matrices (PSSMs) of these domains[133,134], allowing for low-identity hits with e-values < 1:

rpstblastn -query <library.fa> -db proteinNCBICDD1000/selection/Selection1000Library -outfmt "7 stitle evalue qstart qend" -evalue 1 -outlibrary.fa.txt

**Annotation of repeats across the krill genome with RepeatMasker.** We split the genome assembly into 100 chunks and used RepeatMasker v4.1.1 to annotate each chunk in parallel in the high-performance Uppmax computing environment:

RepeatMasker -pa 14 -norna -e rmblast -a -u -gff -xsmall -gccalc -lib <repeat_library.fa>

We used the RepeatMasker tools calcDivergenceFromAlign.pl to calculate the Kimura 2-Parameter divergence metrics from repeat alignments and createRepeatLandscape.pl to generate an overall repeat landscape distribution. We also used a custom script to parse the RepeatMasker.out file and estimate the amount of every TE superfamily (or similar).

Statistics about the repeat library and repeat annotation is available in Supplementary Data 3.

## Gene annotation

We used both RNA data and comparative data from other genome-sequenced crustaceans to annotate genes in the krill genome, with the ultimate aim to produce a non-redundant set of protein-coding genes that could be used for genome and SNP annotation.

**RNA-based data.** We mapped 755 M Illumina short-read RNA-seq read-pairs derived from the reference specimen itself and two published specimens (one male; one female)[94] against the assembly with HISAT2 v2.2.1[109] (mapping rate 88.2%). We then mapped 8.7 M full-length PromethION cDNA reads produced from the reference specimen against the assembly using minimap2 v2.17-r974 (mapping rate 95.8%). StringTie v2.2.0[135] was used in "mixed" mode to generate reference-guided RNA transcript models in GTF-format from the short-read and long-read mappings simultaneously.

Trinity transcripts that had been assembled independently from the genome assembly (see above) were used as a second line of RNA-based evidence of genes. Only putatively coding transcripts with long open reading frames and domain hits against Metazoan templates (n = 60,677) were carried forward in this analysis. SPALN v2.4.6[136] was used to build a DNA database of the assembly (-KD) and to produce spliced alignments of the Trinity transcripts in GFF3 format ("-Q7 -LS -d<genome database > -O0,1,3").

**Comparative data.** We aligned peptide sequences from the KrillDB reference transcriptome from the Antarctic krill *E. superba*[137] and published gene models from six genome-sequenced crustaceans (n = 304,549; Supplementary Table 3) against the genome assembly using SPALN, with the ambition to potentially recover additional gene models not well-represented by our RNA data. A protein database was built for the assembly with SPALN (-KD) and spliced protein alignments were then produced in GFF3 format ("-Q7 -LS -d<genome database > -O0,1,2,3,4"). For downstream compatibility, a custom Perl script was used to ensure that start coordinates for all features were always smaller than stop coordinates, regardless of strand orientation. GFFCOMPARE v0.12.6[138] was used to combine the protein-based GFFs and assign the alignments to a set of common loci with different isoforms. We used TransDecoder v5.5.0 to only keep candidate

alignments that retained significant protein domain homology by: i) converting GTF to GFF3 (gtf_to_alignment_gff3.pl); ii) generating matching sequences in FASTA format (gtf_genome_to_cdna_fasta.pl); and iii) search for long open reading frames (TransDecoder.LongOrfs) in strand specific mode ("-S").

The candidate ORFs were searched for homology against known proteins in the Swissprot database (timestamp 2021-06-16) with BLASTP v2.9.0 +[108] and the Pfam database (release 34.0) using HMMER3 hmmscan v3.3 (http://hmmer.org/), respectively, according to standard procedure (https://github.com/TransDecoder/TransDecoder/wiki). TransDecoder.Predict was used to identify the peptide alignments with significant hits, cdna_alignment_orf_to_genome_orf.pl to write a GFF3 file with the corresponding genome coordinates and GFFREAD[138] to convert in into GTF format. To reduce the risk of carrying forward low-quality protein alignments and with the aim to generate a non-redundant set of gene models, we used custom Perl scripts to parse the GFFCOMPARE locus *.tracking file in order to: i) discard loci labeled by only one crustacean species; and ii) to select one representative alignment from each locus with the highest Swissprot BLASTP score from the preceding step; and iii) generate an updated GTF file.

**Consolidation of gene models.** We combined the three sources of potential gene models generated above (*i*=StringTie RNA-seq transcripts; *ii* = SPALN-aligned Trinity transcripts; *iii*=non-redundant comparative crustacean models) with GFFCOMPARE and predicted coding transcripts with Transdecoder (as in the previous section). We next queried all predicted peptide sequences against the full invertebrate NCBI RefSeq database (timestamp 2021-03-05, release 204) using DIAMOND v0.9.9[139]. From each gene locus identified with GFFCOMPARE, we selected the gene model/isoform with the maximum DIAMOND alignment score to a RefSeq sequence, in order to produce a set of high-quality and non-redundant gene models across the krill genome assembly. The gene labels of both the non-redundant and redundant gene sets were tagged to reflect the source of the evidence, including the best model of each locus.

**Gene set size and completeness.** Altogether, 202,138 models were generated from combining the datasets as above, out of which 118,528 models were found to be putatively protein-coding with TransDecoder. After removing redundancy by selecting the single best DIAMOND +RefSeq model or isoform, 42,227 non-redundant gene models were retained. Of these, 30,766 (73%) models were selected on the basis of a highest-scoring reference specimen StringTie model (assigned "REF_STRG" tag in the sequence header), 8,063 (19%) models from a reference specimen Trinity transcript ("REF_TRIN" tag) and 3,398 (8%) models from comparative peptide data from other crustaceans. TransDecoder annotated genes as "*complete*", "*3prime_partial*" (i.e. missing terminal exons or stop codons), "*5prime_partial*" (missing start codon) or "*internal*" (missing both start and stop codons). 26,448 models were annotated as "*complete*" or "*3prime_partial*" (63%), 7,379 were annotated as "*5prime_partial*" and 8,400 annotated as "*internal*" (Supplementary Data 4). We used a custom Perl script to detect missing in-frame stop codons directly downstream of models annotated as "*3prime_partial*" or "*internal*", restoring stop codons in 5,884 gene models, primarily derived from transcript or peptide data (Supplementary Data 4).

We then used both BUSCO v3.0.2b[91] with the Arthropoda odb9 lineage set and BUSCO v5.0.0[97] with the Arthropoda odb10 lineage set, respectively, to assess the completeness of the gene models (Fig. 1b, Supplementary Fig. 4).

**Functional and evolutionary annotations.** We applied several tools to accomplish functional annotations of the gene sets. First, we used EnTAP v0.10.7[140] with the SwissProt and invertebrate RefSeq peptide databases referred to above. To this base resource, we added gene models from

multiple genome-sequenced arthropod species not represented in our release of RefSeq (Supplementary Table 4). We ran EnTAP in protein-mode ("--runP") with default settings, using the TransDecoder-predicted peptide sequences derived from the krill gene models and EnTAP's pre-configured EggNOG framework resources[141] to annotate genes. This resulted in 27,584 of the 42,227 gene models being annotated, while 14,643 remained unannotated (Supplementary Data 4). Second, we downloaded the *Drosophila* FlyBase database release FB2021_01[142] and used BLASTP v2.9.0 +[108] to find *Drosophila melanogaster* (dmel_r6.38; n = 30,724 peptides) homologs of our krill peptides. For each krill peptide sequence, the best scoring *Drosophila* homologue was retained as a gene label for downstream annotation of genes or SNPs without further resolving best reciprocal relationships. The "FBgn" Flybase IDs were used to compute Gene Ontology enrichments for gene family expansions and for divergence signals between populations (see sections below).

**Gene region masking.** The non-redundant set of protein coding gene was used to generate a mask across the whole genome with a custom Perl script, in which every base was substituted for the type of gene region it occurred in (1=intergenic; 2=intron; 3 = 3′-UTR; 4=exon; 5 = 5′-UTR; 6=cds including start/stop codons). This was used to enumerate the lengths of different gene regions and to classify specific positions (e.g. in the subsequent DNA methylation analyses or assessments of variation across the genome).

We then estimated repeat-content for each gene to test for correlations with annotations (as a control) and expose potential transposon models among unannotated gene models. We therefore used the krill repeat library to mask the coding sequence (CDS) of all gene models with RepeatMasker (using the same settings as the whole-genome masking). We aimed to measure and contrast the repeat content between "regular" gene models that had been annotated for function and homology towards other invertebrate genes using EnTAP against those matching transposable elements in other species or those that were unannotated. To distinguish between regular gene annotations and transposable element annotations, we applied keyword searches against: i) each (per-gene) line in the EnTAP output; and ii) the descriptions matching the COG (Clusters of Orthologous Genes) and eggNOG tags reported by EnTAP (field number 31 "EggNOG Member OGs" in the EnTAP summary table). To match a putative transposon annotation, either the whole line itself or the COG/eggNOG ortho-group descriptions had to:

i. have a match to either of the keywords: "retrotrans" or "transposon" or "transposable" or "transposase or "reverse transcriptase" or "RVT_1";

ii. but not a match to the keywords: "retrotranslocation" or "piRNA" or "miRNA" or "prevent" or "repress", since these keywords were associated with genes responsible for inhibiting transposable element activities.

This resulted in the identification of 25,301 regular protein-coding gene models and 2,283 transposable element annotations and those models that were unannotated in EnTAP (n = 14,643). Their respective lengths, repeat content and other features are provided in Supplementary Data 4 and Supplementary Fig. 6.

### Evolutionary analyses of protein coding genes
We aimed to assess gene and gene family evolution in the krill by comparing the set of coding genes in our annotation to orthologs of nine other crustaceans (Supplementary Table 5). We prepared the data by removing redundant gene sequences from published gene models of the other species, keeping only the isoform with the longest peptide sequence for every gene.

**Inference and comparison of orthologs between krill and other species.** We performed a search for orthologs using ProteinOrtho v6.0.14[143] with DIAMOND v0.9.30. We used the 26,448 non-redundant

Northern krill gene models that had been annotated as complete in this analysis. We then enumerated all 1:1 single-copy ortholog pairs between the krill and the other species. We used 1:1 orthologs between the krill and those of *H. americanus*, *P. monodon*, *H. azteca* and *E. affinis*, respectively, for direct comparative analysis of the lengths of coding sequence and full gene bodies in R[144]. Gene length comparisons are provided in Fig. 1., Supplementary Fig. 7 and Supplementary Data 5. For the Antarctic krill, comparisons were done against the mean length reported in the Antarctic krill genome paper[24] (no gene models were available at the time of preparing these analyses).

**Estimating divergence between the Northern krill and the Antarctic krill.** We estimated per-base divergence at synonymous sites (*dS*) between the Northern krill and the Antarctic krill as a proxy for nearly neutral divergence comparably unaffected by selection. We then used this estimate to date when the two species split from their most recent common ancestor.

To this end, we first used DIAMOND to detect 13,371 pairwise Reciprocal Best Hits between 140 K Antarctic krill transcriptome protein models from the KrillDB2 and 26,395 non-redundant Northern krill protein coding gene models (see section below about the gene models). RBH protein sequences were aligned with MAFFT v7.453[145] using settings recommended to avoid over-aligning regions with poor homology[146]:

mafft --globalpair --allowshift --unalignlevel 0.8 --leavegappyregion --anysymbol seq.aa.fasta > seq.aa.fasta.ginsi.fasta

For each alignment, we fitted the corresponding nucleotide sequences with PAL2NAL v14[147] and then used KaKs_Calculator v1.2[148] with the approximate "YN" method to estimate synonymous and non-synonymous sites and substitutions and compute *dN*, *dS* and *dN/dS*, while accounting for unequal base frequencies, transition/transversion rates and multiple substitutions between sequences[149]. We summed the total synonymous sites and substitutions to derive a single overall *dS* estimate across all 13,371 genes.

In addition, we concatenated the genes and computed the observed and Jukes-Cantor-corrected distances between the two species across full sequences or 1st, 2nd and 3rd positions separately using Seaview v5.0.5[150].

Using the *dS* estimate of 0.46, we next estimated divergence time assuming the mutation rate inferred from snapping shrimp in ref. 30 of 2.64e-9 substitutions per site per generation and a 1.5 year generation time (1 year in Northern krill and 2 years in the Antarctic krill)[9,151]. We used a standard linear equation assuming a constant molecular clock:

$$T = \frac{D * g}{2u} \quad (1)$$

Here, *D*=divergence per base (*dS* = 0.46), *g*=generation time (1.5), *u*=mutation rate (2.64e$^{-9}$).

Results of these calculations are provided in Supplementary Data 5.

**Inferences of phylogenetic interrelationships.** We detected 1,011 single-copy orthologs (OGs) across all ten species and used these genes to produce a time-calibrated phylogenomic crustacean species tree. We first aligned the peptide sequences of each OG with MAFFT and filtered the alignments with Gblocks[152]:

Gblocks <OG.fa > -t = p -b1 = 6 -b2 = 6 -b4 = 5 -b5 = h -b6 = y -v = 10000 -d = y -e = .gb

We then inferred phylogenetic interrelationships with IQ-TREE v2.1.0[153], applying the best-fit LG + F + I + G4 model (Modelfinder+BIC) across all filtered ortholog alignments as a supermatrix and running 1,000 ultrafast bootstrap replicates[154]. The crustacean phylogeny was converted into an ultrametric chronogram using r8s v1.8.1[155]. Input for

r8s was generated using a helper script from the CAFE v5.0[156], where we specified that the phylogeny had been inferred from 522,746 alignment sites. We used three calibration points, setting: i) the oldest split in the tree (between *D. magna* and all others) to have occurred 530 M years ago[157]; ii) the split between the lobster *H. americanus* and the crayfish *C. quadricarinatus* to be at least 372 M years old (i.e. older than the radiation of lobsters) and the split between krill and decapods to be at least 447 M years (i.e. older than the radiation of decapods) according to molecular dating in[158]. We ran r8s using the penalized likelihood method:

divtime method=pl algorithm=tn cvStart=0 cvInc=0.5 cvNum=8 crossv=yes

**Analysis of gene family evolution.** We used SwiftOrtho (https://github.com/Rinoahu/SwiftOrtho)[159] to broadly cluster genes into putative gene families and used CAFE to trace the expansion or contraction of these families along the time calibrated ultrametric tree inferred in the previous step. We then used wgd[160] to test for signatures of divergence between genes that could indicate a history of whole genome duplication.

**Preparing the krill gene set.** The full krill gene set spanned n = 42,227 putative gene models, out of which n = 27,584 genes had been annotated with EnTAP and 18,962 genes had been tagged as complete using TransDecoder (Supplementary Data 4). Because not all gene models were recognised for function or homology with EnTAP or complete with TransDecoder, the full gene set could contain spurious gene models (e.g. non-coding RNA, pseudogenes or transposable elements) or incomplete gene models (e.g. fragments of the same gene locus separated on unscaffolded contigs) that could artificially inflate gene numbers and estimation of gene family expansions.

From the set of annotated genes, we therefore first removed 2,283 potential transposable elements (TEs) that had been annotated as TEs using EnTAP, leaving 25,301 models in this set. We then performed a relaxed homology search between unannotated krill genes and the nine crustaceans (Supplementary Table 5) with DIAMOND using an e-value threshold of 1e-5 and detected significant hits for 1,934 hitherto unannotated genes. These genes were added to the annotated set for a total of 27,235 genes. This set was then checked for potentially fragmented models that may belong to the same underlying gene locus through queries against two contiguous sets of RNA sequences:

a. We mapped the gene sequences back to the Trinity transcripts (using the longest isoform of each putative Trinity gene) to identify genes that mapped to different parts of the transcripts. For each gene, the RNA transcript with the longest total BLAST hits was registered. If gene models mapped to the same transcript and their respective alignments did not overlap at all or only overlapped marginally (at most 10% of the length of the gene), they were taken potential separate parts of the same gene body in the genome. In these cases, the shorter and potentially redundant gene fragment(s) were removed from the set, so that only one representative gene model was kept.

b. We also mapped the gene models to the full length Nanopore cDNA sequences that had previously been clustered with VSEARCH (see above; n = 25,484), and detected short and potentially redundant fragments of the same underlying gene using the same approach.

This procedure identified and removed 839 gene fragments from the gene set, the majority of which had been marked as incomplete by TransDecoder, resulting in a final krill gene set spanning 26,395 non-redundant gene models that were carried forward for gene family analysis.

**Analyses with SwiftOrtho to cluster genes into families.** SwiftOrtho inference of gene families was performed in three steps for each dataset:

i. find similar matches between sequences in the non-redundant crustacean gene sets:

find_hit.py -p blastp -i <species.fa > -d all_proteins.fa -a 20 -e 1e-5 -s 111111 -o <species.fa.out>

ii. infer potential orthology relationships between genes using low taxon coverage ("-c 0.3") and identity thresholds ("-y 30"):

find_orth.py -i <all_matches.out > -a 16 -c 0.3 -y 30 > <all_matches.out.orth>

iii. cluster similar orthologs or paralogs into groups/gene families using the Affinity Propagation Cluster (APC) algorithm:

find_cluster.py -i <all_matches.out.orth > -t 16 -a apc -I 1.5 > <all_matches.out.30_30.orth.apc>

The clustering resulted in a gene count spanning 17,119 gene families with sequences from two or more species. This set included 11,259 families with at least one gene from the krill and 6,141 families where both the krill and the outgroup *D. magna* were represented. For comparison, the lobster *H. americanus* and amphipod *H. azteca* were represented in 12,274 and 9,149 gene families, respectively.

**Analyses with CAFE to trace gene family expansions or contractions.** The SwiftOrtho gene family dataset was reformatted into the expected input for CAFE using a custom Perl script and analyzed in CAFE5. The software uses a maximum-likelihood framework to estimate the rates and events of gene family evolution using a birth-death process to model the gain or loss of genes along the branches of the species tree[156]. These analyses are conditioned on there being at least one gene present in the outgroup taxon for a gene family to be included. Our analysis of gene family evolution was therefore restricted to families with at least one representative in the *D. magna* (n = 6,814).

Here we followed the steps outlined by the developers in https://github.com/hahnlab/CAFE5/blob/master/docs/tutorial/tutorial.md:

1. Separating extremely large gene families from the rest to reduce variance in gene copy numbers and improve parameter estimation:

python clade_and_size_filter.py -i <all_matches.out.30_30.orth.apc.table.ALL.csv > -o all_matches.out.30_30.orth.apc.table.ALL.csv.filtered -s

In total, 16 large families were separated from the rest in the first size filter step, all of which indicated expansion in other species than the krill. For this reason, those 16 families were not further analyzed.

2. Infer the gene family evolutionary rates (the so-called lambda parameter) and compute gene family evolution across the species tree.

cafe5 -i all_matches.out.30_30.orth.apc.table.ALL.csv.filtered -t filtered.iqtree.out.contree.rooted.nwk.r8s_ctl.txt.out.tre

We compiled distributions of gene family sizes for each species, estimated the optimal lambda across the tree to be 0.0005932 and inferred the patterns of expansion/contractions across the 6,814 gene families. Results are provided in Fig. 1, Supplementary Fig. 7 and Supplementary Data 6.

**Gene ontology (GO) enrichment analysis of expanded gene families.** The subset of gene families with significant evidence of expansion in the Northern krill was analyzed for function by assessing the corresponding *D. melanogaster* homologs using ontologies in the online Flybase service. Using the same *Drosophila* homologs, we tested for enrichment of Gene Ontology terms among the expanded families vs the non-expanded ones using the online services GOrilla[161] and ShinyGO (v0.77)[162] using False Discovery Rates (FDR) thresholds of 0.05

after correcting for multiple testing. Analyses of gene families were carried out in two ways with regards to the *Drosophila* homologs: i) using all homologs associated with a gene family (which may be highly variable in the number of homologs between gene families); ii) using only the most commonly detected homolog for every gene family (only using more than one in case of ties; an approach to reduce variance in the numbers of homologs used to characterize each gene family).

The results of these analyses are presented in Fig. 1 and Supplementary Data 6.

**Analyses with wgd to test for signatures of whole-genome duplication (WGD).** The amount of divergence at neutral sites between two homologous sequences can serve as a proxy for the time that has passed since they originated through duplication. In the case of genes, such divergence can be measured as synonymous substitutions per synonymous site in pairwise alignments between two sequences ($K_S$), assuming that such sites evolve neutrally. Genes can be added to the genome through small-scale duplication events at different points in time or through large-scale duplications including whole-genome duplications, in which many genes originate at the same time. In each scenario, gene duplicates are often lost after some time. The former process can be expected to produce an L-shaped exponential $K_S$-distribution among surviving paralogues in the paranome, suggesting many observed paralogues are young, while the latter process can produce one or more characteristic secondary peaks along the $K_S$-distribution, indicating that those paralogues originated at a particular event and age[163]. To study these patterns, we analyzed the gene set of the krill and, for comparison, those of five other crustaceans that have not been associated with whole-genome duplications (*H. americanus*; *P. monodon*; *H. azteca*; *E. affinis*; *D. magna*) with wgd[160]. The tool internally uses DIAMOND to detect similarity among sequences, MCL[164] to cluster them into putative gene families, FastTree to produce per-family gene trees[165], PAML[166] to perform model-based estimation of synonymous divergence between paralogs and Scikit-learn[167] to fit mixtures of components to the overall $K_S$-distribution in order to find evidence for mixed distributions that could indicate WGD. For each species, we:

1. Performed clustering of the genes:

singularity exec wgd.sif wgd dmd --nostrictcds <cds.fasta>

2. Computed and visualized the respective $K_S$-distribution:

singularity exec wgd.sif wgd ksd -n 16 --preserve --max_pairwise 250000 wgd_dmd/<cds.fasta.mcl > <cds.fasta>

singularity exec wgd.sif wgd viz -ht barstacked -r 0 5 -b 50 --weighted -ks <cds.fasta.ks.tsv > -o <cds.fasta.ks.tsv.viz.svg>

For the krill, we then performed additional analyses using Gaussian Mixture Models and applying models with 1 to 4 components and applying the Bayesian Information Criterion to test which model best explained the data:

singularity exec wgd.sif wgd mix --method gmm -n 1 4 -o wgd_mix_gmm <cds.fasta.ks.tsv>

The results of these analyses are presented in Supplementary Fig. 8 and Supplementary Data 6.

**Characterization of the opsin repertoire.** We used BLASTP to query the full protein-coding gene set of the Northern krill (n = 42,227) and the Antarctic krill *E. superba* transcript set from Urso *et al*.[32]. (n = 151,585) against the crustacean opsin gene family sequences from Palecanda *et al*[34]. (n = 631). Sequences with hits spanning ≥50 amino acids and that had ≥50% identity were taken forward for analysis. We reduced redundancy among the Antarctic krill sequences by running CD-HIT ("-c 0.85 -n 5 -M 0 -T 40"). We used MAFFT with the "--add" parameter to align the sequences (19 from *M. norvegica* and 15 from *E. superba*) to the Palecanda alignment and FastTree was used with the WAG model to produce a gene family tree spanning

663 sequences (Supplementary Fig. 9). A tree containing only krill sequences (n = 65) was produced by pruning all other sequences (Supplementary Fig. 10).

**Characterization of the Hox gene complement.** The presence and number of duplicate Hox genes can indicate ancestral whole genome duplication in a lineage, as these genes are often retained in duplicate copies even after diploidization[168]. To test for duplicated Hox genes in the krill, we used curated Hox homeodomain sequences from a previous analysis[169], HomeoDB2[170] or NCBI. We searched the krill genes against 10 core homeodomain sets from crustaceans and selected insects, spanning 60–80 amino acids and 3–6 species per set, to annotate Hox genes and detect Hox gene duplication. Eight well-curated homeodomain sets were used unchanged from analyses of the amphipod *P. hawaiensis*[169] (*lab=labial*, *pb=proboscipedia*; *Hox3*; *Dfd=Deformed*; *Scr=Sex combs reduced*; *Antp=Antennapedia*; *abd-A=abdominal-A*; *Abd-B=Abdominal-B*), one set was downloaded from HomeoDB2[170] (*ftz=Fushi Tarazu*; HomeoDB2 species: *D. melanogaster* [Dm]; *Apis mellifera* [Am]; *Tribolium castaneum* [Tc]) and the final set was derived from both HomeoDB2 and NCBI sequences (*Ubx/Utx=Ultrabithorax*; HomeoDB2 species: *D. melanogaster* [Dm]; *A. mellifera* [Am]; *T. castaneum* [Tc]; NCBI species: *P. hawaiensis* [Ph] FJ628448; *D. magna* [Dmag] BAE96992.1). Supplementary Data 7 lists all homeobox reference sequences used.

We queried the krill gene against the 10 homeodomain sets using BLASTP. We kept the longest matching peptide substring (the putative homeodomain) for sequences that had at least one hit to a homeodomain template which spanned ≥50 amino acids and had ≥50% identity (n = 119 sequences). Because the homeodomain is relatively similar across the different genes, we then combined all 10 sets and krill substrings into a single FASTA sequence file and aligned all data at once with MAFFT. We then inferred a homeodomain gene family tree with Fasttree v2.1.11[165] and inspected the phylogenetic positions of the candidate krill genes.

The Hox gene family tree is shown in Supplementary Fig. 11.

**Characterization of genes involved in DNA methylation.** The presence or absence of genes that encode enzymes involved in DNA methylation (i.e. DNMT1 or DNMT3), DNA demethylation or associated damage repair (e.g. TET family; ALKB2) vary among arthropods and may indicate whether the DNA methylation pathway is operational[37]. Since we had detected DNA evidence of genome-wide DNA methylation through Nanopore signal analysis (see below), we searched the krill gene-set for homologs encoding the DNMT1/2/3, TET2 and ALKB2 enzymes. We first identified potential candidates using the gene names of best-matching genes from the EnTAP gene annotation. We then combined arthropod sequences from multiple sources with established or likely orthology, including OrthoDB v10.1[171], EggNOG v5.0[172] and NCBI (mostly crustacean sequences) (Supplementary Data 7) and performed phylogenetic analyses of the position the candidate krill genes in each gene tree. For each gene, we aligned the peptide sequences with MAFFT, trimmed poorly aligned regions with trimAl v1.2rev59[173], inferred a gene tree with FastTree and inspected the position and the length of the branch at which the krill sequence(s) grouped in the tree. For further validation, we performed analyses using the online NCBI Conserved Domain Search[133,134] and SWISS-MODEL[174] for homology-based inference of structure and domains in each candidate sequence. SWISS-MODEL uses a comprehensive search strategy to detect significant matches, including the composite Qualitative Model Energy ANalysis (QMEAN) scoring method to analyze and score protein structures.

**Predicting protein structures.** To further assess the accuracy of these candidate proteins possibly involved in the DNA methylation system, we predicted their three-dimensional protein structure. The protein

sequences were added to the ColabFold server[175], where the input multiple sequence alignment is built by a homology search by MMseq2[176] before modeling by AlphaFold2[177]. No PDB templates were added, and 3 recycles were used. Five models were generated per protein.

## Analysis of DNA methylation levels

Nanopore signals are altered by DNA base-level modifications, including 5-methylcytosine (5-mC) at CpG sites, that can be detected in silico using Nanopore data[178]. We scanned the Nanopore reads for evidence of methylated cytosines across all well-covered CpG sites in the nuclear and mitochondrial genomes (i.e. DNA methylation in the somatic muscle tissue that the sequences were derived from), and cross-referenced methylation frequencies with genomic features.

We first mapped the PromethION long-reads back to the finished krill genome assembly using minimap2 as above and then used the GPU-accelerated program f5c v0.6[179] to scan for evidence of methylated cytosines across all CpG sites in the genome with 10× or higher read coverage. We used three f5c subcommands:

1. "index" to index the raw FAST5 PromethION signal data and reads;
2. "call-methylation" to call the likelihoods for CpGs to be methylated or not;
3. "meth-freq" to compute the frequency of reads with sufficient evidence of methylation at every CpG.

We then used a custom Perl script to group the reported methylation frequencies according to genomic regions using a genome mask (see above). Depending on how closely spaced CpG sites are, f5c sometimes outputs joint methylation frequencies across a short subsequence with more than one CpG site. In such cases, we split the observation into separate data points with identical methylation frequencies and queried each position against its corresponding genomic region. We then compiled a second round of summary statistics removing potential source of error by: i) masking all CpG sites at which we had observed heterozygous genotypes in the reference specimen; ii) excluding all regions that were marked inaccessible due to high or low depths in the population genomic dataset (see below). We visualized the results in R and computed mean, median and 95% confidence intervals (200 bootstrap replicates; two-tailed comparisons).

We cross-referenced the DNA methylation data with information about reads and repeats. First, for every gene model we counted the number of RNA splice isoforms inferred from RNA-seq data and StringTie (see above), keeping genes supported by at least one isoform (n = 21,031). We partitioned genes into 5%-bins of average DNA-methylation level and computed the mean number of isoforms in each bin, along with 95% confidence intervals (1,000 bootstrap replicates; two-tailed comparisons). We then extracted 1,706 putative LTR retrotransposons originally reported by LTR_Retriever and verified to contain at least one expected LTR domain when queried against the GyDB2 database (as above). Identities between 5'-LTR and 3'-LTR regions estimated by LTR_Retriever were used to partition the LTRs in bins of 5% divergence (here 0 to low divergence between LTR regions is taken as a proxy for being an evolutionarily young and recently inserted repeat).

Results are summarized in Fig. 1 and Supplementary Fig. 17.

## Population genetic data processing and analyses

**Processing resequencing data.** The population-scale data was mapped to the genome, read-group tagged and marked for optical duplicates using standard NGStools. We made a per-base depth profile across the genome to mark positions beyond ±50% of an observed putative diploid peak (Supplementary Fig. 18A) as inaccessible for measuring variation. SNPs and short structural variants

were then called, decomposed and quality-filtered for annotation and analyses.

**Mapping, read-group tagging and duplicate marking datasets.** The barcoded short-read resequencing data was demultiplexed by NGI, resulting in one or more paired FASTQ archives per specimen (one pair per lane). The genome assembly was indexed using BWA as above. For each sample, we mapped and readgroup-tagged the data using BWA v0.7.17 with the "mem" algorithm and sorted the output on the fly using samtools v1.12:

bwa mem -t 18 -R <read_group > <reference > <fwd.fastq.gz > < rev.fastq.gz> | samtools sort -@ 4 -O bam -m 4 G -o <mapped_sample > . bwa.bam -

BAM files for samples sequenced on multiple lanes were merged with samtools. The merged data of each sample was then sorted by read name, marked for duplicates using samblaster v0.1.24[180] and again re-sorted by genome coordinates:

samtools sort -@ 10 -n -O SAM <mapped_sample.bam > | samblaster | samtools sort -@ 10 -O BAM -o <mapped_marked_sorted.bam>

For the reference specimen, both the high-coverage linked-reads and low-coverage data were used. We mapped the high-coverage linked-reads and processed it in the same way. All BAMs were indexed with samtools.

**Mapping-depth profiles and masking of inaccessible sites.** Our analyses of repeats had indicated that the krill genome assembly was highly repeated (see above) and that short-read mappings tended to result in uneven coverage across the genome, suggesting that many genomic regions could be problematic for SNP-calling. We therefore produced depth of coverage profiles across the genome with the aim to mark regions deemed to be accessible or inaccessible for SNP discovery. To this end, we compiled a per-base depth-track across the whole genome for each specimen, counting reads that mapped with a minimum quality of 10:

samtools view -q 10 -b <sample.bam > -b 'cat contig.list' | samtools depth -a --reference <genome_assembly.fa>

Using custom Perl scripts, we merged these depth tracks into per-population tracks, as well as one overall depth-track for the whole dataset. We compiled a depth distribution across the whole genome and identified a peak of coverage at ~188× and set lower and upper thresholds to be ±50% (94×) around this peak (Supplementary Fig. 16). Positions where less than 50% of samples were mapped (n < 37) were masked as inaccessible. Downstream, we only considered SNPs and estimated patterns of genetic variation from genomic regions with 94–281× depths of coverage (8.43 Gbp) and for which data had been mapped in at least 50% of samples (n = 37). We produced a per-base genome mask marking sites as either accessible ("1") or inaccessible ("0") and used this mask to correct estimates of the levels of genetic variation in the genome. The accessibility mask was cross-referenced against the mask of genome regions (see above).

**Calling and phasing single-nucleotide polymorphisms (SNPs) across the genome**

**Calling and processing variants.** We subdivided the genome sequences (scaffolds and unscaffolded contigs) into 160 approximately equally sized chunks specified GTF files to enable SNP-calling different parts of the genome in parallel. We then used FreeBayes v1.3.4[181] to call variants with a theta prior set to "-T 0.01", "--use-best-n-alleles 5" to limit memory usage, "-m 10" to limit analysis to read alignments with mapping scores above 10 (matching depth estimates in a previous step), "-q 20" as a minimum base quality filter for alleles and "-E 0" to limit the generation of complex haplotype variants. We also used inclusive depth filters outside mapping depth thresholds: "--min-coverage 80" as a lower threshold and "-g 474" as an upper

threshold and piped the output to the Vcflib v1.0.1[182] vcffilter tool set to only keep variants with QUAL > 20:

freebayes -T 0.01 -E 0 -m 10 -q 20 --min-coverage 80 -g 474 --use-best-n-alleles 5 -t <genome_part_${NUM}.gtf > -f <reference.fasta > < bam files> | vcffilter -f \"QUAL > 20\" > <variants_${NUM}.vcf>

We processed the called variants to decompose complex haplotypes and only keep biallelic SNPs. First, we used bcftools v1.12[101] with the "norm -d all -a" parameters to left-align and normalize indels, decompose complex variants and only keep one record per position. The output was piped to vt v0.5772[183] to split multiallelic variants while correcting for read counts ("-s"):

cat variants_${NUM}.vcf | bcftools norm -d all -a -f <reference.fasta > | vt decompose -s - > <variants_${NUM}.decomposed.vcf>

We then used two custom Perl scripts to:

i. remove variants that fell outside accessible regions
vcf2filtered_vcf_by_coverage.pl --vcf <variants_${NUM}.decomposed.vcf > --coverage <depth_mask.fasta > --seqs <genome_part_${NUM}.gtf > --verbose

ii. keep only biallelic SNPs in which at least 50% of samples had been genotyped and while also ensuring that these SNPs occurred within the desired thresholds:
vcf_biallelic2fasta.pl --input variants_${NUM}.decomposed.accessible.vcf --output variants_${NUM}.decomposed.accessible.finished.vcf --min_fill_position 0.5 --min_depth 94 --max_depth 281

**Imputing and phasing SNPs.** Each chunk of SNPs (n = 160) was imputed and phased with BEAGLE v4.0 r1399[184] in a two-step process. We first imputed missing genotypes using genotype likelihoods:

java -Xmx48g -jar beagle.27Jan18.7e1.jar nthreads=10 gl=variants_${NUM}.decomposed.accessible.finished.vcf out=variants_${NUM}.decomposed.accessible.finished.imputed.vcf

We then phased the data using the inferred genotypes:

java -Xmx48g -jar beagle.27Jan18.7e1.jar nthreads=10 gt=variants_${NUM}.decomposed.accessible.finished.imputed.vcf.gz out=variants_${NUM}.decomposed.accessible.finished.imputed.phased.vcf.gz

This resulted in a dataset spanning ~760 million biallelic SNPs.

**Annotating SNPs.** We used GFFREAD to extract the CDS DNA sequences of the non-redundant protein-coding genes (see gene annotation above):

gffread -g <genome.fasta > -x <cds.fasta > <genes.gff>

The GFF gene coordinate file and genome and cds sequences were then used to build a custom database with SnpEff 5.0e[185]:

java -Xmx96G -jar snpEff.jar build -gff3 -v mnor 2 > &1 | tee build.log

Each of the 160 phased VCF files was then annotated with SnpEff, generating a new compressed VCF:

java -Xmx16g -jar snpEff.jar -v mnor <phased.vcf.gz > -csvStats <phased.annotated.vcf.gz.summary.tsv > -htmlStats <phased.annotated.vcf.gz.summary.html > | pigz -p 4 -c - > <phased.annotated.vcf.gz>

**Estimating patterns of variation**

**Levels of variation.** From the SNPs, we computed the per-base pair population mutation parameter Watterson's theta ($\theta_W$; the number of segregating sites) and nucleotide diversity ($\pi$; the average number of pairwise nucleotide differences between a pair of chromosomes sampled from a population) as estimates of genetic variation across the whole population dataset and for each of the eight populations separately. These statistics were computed across non-overlapping windows of 1,000 bp or 100,000 bp, or across whole sequences. The effective length of each window was corrected for the number of actual accessible bases according to sequence depth (see above) and SNPs and accessible bases were further subdivided according to intergenic, UTRs, cds, synonymous/non-synonymous coding positions or intronic sequence to enable estimation of variation across different kinds of

genomic regions. The 1 kb window estimates were used to map variation stepping away from genes, while the 100 kb estimates were used to compile genome-average statistics. In addition, we re-used code from BioPerl[186] to compute Tajima's D[187] by comparing $\theta_W$ and $\pi$, to characterize the degree to which the genome overall appeared to evolve neutrally ($D_T \approx 0$) or depart from neutrality ($D_T \neq 0$), which may indicate effects of population bottlenecks or selection.

Synonymous and non-synonymous variants were enumerated from the SnpEff annotations while the number of synonymous and non-synonymous sites across gene bodies were estimated using KaKs_Calculator v1.2[148]. As in the window-based analyses, a correction was made for the per-base diversity estimate to take into account the number of accessible sites in the coding sequences.

**Effective population size ($N_E$) and its historical demographic trends.** We inferred long-term $N_E$ under mutation-drift equilibrium for an idealized population, assuming a snapping shrimp mutation rate of 2.64e$^{-9}$ substitutions per site and generation[30]. This is to the best of our knowledge, the closest malacostracan species with a robust estimate of the mutation rate. We then used the standard equation to compute $N_E$ using the genome-wide average value for $\theta_W$:

$$N_E = \theta_W / 4\mu \qquad (2)$$

We inferred past changes in $N_E$ by estimating haplotype coalescent times using patterns of heterozygous genotypes in the reference specimen alone using two implementations of the Pairwise Sequentially Markovian Coalescent. In each case, we restricted analyses to span the distribution of heterozygosity across scaffolds or contigs longer than 500 kb, reusing the same SNP calls and masks of accessible sites (see above) as used for the full dataset.

I. PSMC[40]: we first applied the Pairwise Sequentially Markovian Coalescent (PSMC) as implemented in psmc v0.6.5-r67 https://github.com/lh3/psmc (n = 4,911 scaffolds; 3.48 Gbp). Using a custom Perl script, we converted our depth of coverage genome mask to the window-based FASTA-like input format used in PSMC. To accommodate the high levels of variation in the krill, we used a window size of 10 bp instead of the default of 100 bp. We encoded each window with at least one accessible base with the symbol "T" (n = 156,739,078), while fully inaccessible windows were encoded as "N" (n = 169,545,821), and used the VCF files to re-code windows with heterozygous genotypes as "K" (n = 21,271,007). We then applied the PSMC splitfa tool to split long sequences into fragments and ran 100 bootstrap replicates of PSMC, randomly resampling fragments in each replicate. We used a set of time segments with 12 free segments close to the present and ran each replicate for 25 iterations ("-N25"):

for NUM in {1..100}; do psmc -N25 -t15 -r5 -b -p "1+1+10*1+15*2+4+6" -o round-${NUM} <psmcfa.split> done

II. MSMC[188]: we then applied the Multiple Sequentially Markovian Coalescent (MSMC) algorithm for one sample as implemented in mcmc2 v2.1.1 (https://github.com/stschiff/msmc2) (n = 5,176 scaffolds; 3.63 Gb). For a single diploid sample, this method is similar to the original PSMC method above[41]. First, we used a custom Perl script to scan the VCF files and our depth of coverage genome mask to generate the expected genotype input format while correcting for inaccessible sites between SNPs. We then ran MSMC2 with a fine-grained set of time segments:

msmc2 -t 40 -o <out> -p "10*2+100*1+1*2+1*3" <datasets/ *.500kbp>

Alternative time series yielded similar profiles for $N_E$.

In both analyses, $N_E$ and the number of generations are re-scaled post-analysis by the per-generation mutation rate $\mu$, which is unknown for this species. We therefore again applied the mutation rate inferred from snapping shrimp in[30] of 2.64e$^{-9}$ substitutions per site per

generation to scale the statistics, assuming a generation time of one year. For PSMC, we concatenated the results as per the online instructions and plotted the variation among the replicates with the PSMC tool psmc_plot.pl, allowing the program to auto-select the best-fitting iteration for each replicate

psmc_plot.pl -Y 300 -X 5000000 -p -s 10 -u 2.64e-09 -g 1 round-ALL.plot round-ALL

We combined the output from both PSMC and MSMC into a single figure. In this figure we also incorporated the "LR04" benthic $\delta^{18}O$ foraminiferal calcite isotope stack, a record of data that indicates changes in global ice volume and deep ocean temperature across 5.3 million years[42]. This data was downloaded from: https://lorraine-lisiecki.com/stack.html

In addition to these long-term and historical estimates of $N_E$, we interrogated the patterns of linkage disequilibrium (LD) among SNPs to infer effective population size in the more recent past. The spectrum of LD between SNPs at different genetic distances is influenced by recent variation in $N_E$. We therefore analyzed patterns of LD between SNPs along the 199 longest scaffolds using GONE[189], which implements a genetic algorithm to estimate the recent demographic history and $N_E$ that best fits the observed data. We configured the program to treat SNPs as phased ("PHASE = 1") and to use a fixed recombination rate of 0.32 cM/Mb ("cMMb=0.32") (see iSMC estimation below). We analyzed all data together (74 samples), as well as Northern, Eastern and Western subsets of the Atlantic dataset separately, in order to test for variation among different geographic regions.

**Counting alleles and estimating allele frequency divergence, population structure and selection.** Our SNP dataset spanned Northern krill samples collected from eight geographical regions across its natural range (Fig. 1; Supplementary Data 1). We used a custom Perl script to compute allele frequencies at each SNP for each population and across the whole dataset using the genotypes in the VCF GT field in the phased VCF files. The allele counts and frequencies were saved in tabular TSV text files to enable fast access in downstream analyses and used to compute the folded allele frequency spectrum across the whole dataset. We then estimated the pairwise genetic distances and interrelationships between all samples and populations (i.e. the $d_{XY}$ statistic), while correcting for accessible sites. Pairwise distances were converted into a neighbor-joining tree using SplitsTree[190]. We used all detected synonymous polymorphisms (1.3 M SNPs across 7 M sites) to estimate the net synonymous divergence ($D_a$) between the Atlantic (at) and Mediterranean (me) population samples. Here:

$$D_a = d_{xy} - \left( \pi_{S(at)} + \pi_{S(me)} \right) / 2 \qquad (3)$$

The statistic measures residual divergence after correcting for within-group diversity and can be used to determine where along the speciation continuum two populations or presumptive species may be positioned. Generally, $D_a > 2\%$ is likely to indicate reproductive isolation with strong barriers to gene flow[43].

Population structure and ancestry was inferred from the variation among samples using unsupervised PCA and admixture tools, without prior partitioning of samples. We first employed a custom script to subsample one percent of the variant sites, and then pruned remaining variants based on linkage disequilibrium to remove correlation between variant sites in windows of 500 sites. Pruning was done with the function sgkit.ld_prune (https://pystatgen.github.io/sgkit, version 0.5.1) that generates a maximally independent set of variants. We used an $R^2$ threshold of 0.1, whereby no variant pairs below the threshold were retained. PLINK v1.90b4.9[191] was used to perform a PCA on the filtered variants with options '--pca var-wts --double-id --chr-set 46' whose output was used in a custom plotting function in python to generate plots. The filtered variant set was converted to PLINK binary

biallelic genotype table (bed) format to use as input to ADMIXTURE v1.3.0[192]. We varied the number of populations K from 2 to 8 and ran 50 repetitions using different seeds to control initial conditions. We collected the output files (suffix.Q) and uploaded zip archives to the CLUMPAK server (http://clumpak.tau.ac.il/)[193] to summarize the output from the 50 runs.

Specimens were next grouped according to their regions of origin. We used the $F_{ST}$ estimator by Reynolds *et al.*[194] to estimate pairwise allelic divergence across whole scaffolds or contigs and infer the genome-wide levels of divergence among all of the eight populations. The pairwise distances were converted into a NeighborNet network using SplitsTree. We next partitioned the data into two major contrasts representing krill from different regions:

1. The Atlantic Ocean (n = 67 samples) vs. the Mediterranean Sea (Catalan Sea; n = 7 samples).
2. The North Eastern North Atlantic Ocean (samples from waters around Iceland, the Barents Sea, Svalbard and Scandinavia; n = 47 samples) vs. the South Western North Atlantic Ocean (samples from the Gulf St. Lawrence, Canada; and the Gulf of Maine, USA; n = 20 samples)

For each contrast, Reynolds $F_{ST}$ was used to estimate divergence across non-overlapping windows of 100 bp, 1,000 bp or 100,000 bp, whole contigs/scaffolds or at the genome-scale, using a custom Perl script. We used the 1 kbp window-based $F_{ST}$-estimates to compare divergence between genes and flanking regions, and test for association with gene-sequence or selective sweep signals (see below).

The Weir-Cockerham estimator[195] was used to calculate the per-SNP $F_{ST}$, in order to visualize data and outlier loci and SNPs with unusually high levels of divergence. Outlier SNPs were also used to identify gene-level haplotypes of putatively selected gene-variants that segregated between ocean basins. For every contrast or population, the frequency of each haplotype was computed.

We performed a test for isolation by distance, correlating pairwise genetic and physical distance between populations. We first downsampled all populations to the same sample size (n = 7, i.e. limited by the Spanish sample), in order to avoid effects of sample size variation among comparisons and re-estimated genome-wide Reynolds $F_{ST}$ among all populations. We then linearized the genome-wide Reynolds $F_{ST}$ using Eq. (4)[196]:

$$F_{ST} = F_{ST}/(1 - F_{ST}) \qquad (4)$$

For physical distances, we generated linear vectors in Google Earth and traced the shortest path between populations along them but did not consider other oceanographic conditions such as sea currents. We then performed a linear regression to test the relationship between genetic and physical distances using the lm() function in R.

Results of these analyses are given in Figs. 3–4 and Supplementary Figs. 19–23 and 12–26.

**Simulations of divergence.** We performed coalescent simulations of neutral divergence under a simple population-split model using ms[197] to determine the probability of observing high levels of allelic divergence between basin-scale population samples from neutrality alone, in the absence of natural selection. In this Wright-Fisher model, populations would split without any subsequent exchange of alleles through gene flow and diverge over time assuming constant population size and no recombination.

For each of the two major contrasts (see previous section), we parameterized the simulation with the observed genome-wide estimates of divergence and variation. We first inferred the scaled time $T$ by applying Eq. (5)[194] and inserted the value of $T$ into Eq. (6) together with our genome-wide estimation of $N_E$ (1.53 million) to obtain the

number of generations $t$ since the population split:

$$T = \frac{-ln(1 - F_{ST})}{2} \qquad (5)$$

$$t = T * 4N_e \qquad (6)$$

The downsampling of data (see previous section) resulted in 7.35 M SNPs sampled across 8.43 Gb accessible sites across the genome, for an average block length of 1,146 bp per SNP. We therefore aimed to simulate the coalescent process in about 7.35 M 1.1 kbp loci per contrast (making minor adjustments in the case of invariant sites in any of the two contrasts) and export one SNP per locus. Our estimate of population mutation parameter Watterson's theta ($\theta_W$) was 1.62% per base, resulting in a $\theta_W$ estimate of 18.57 per 1,146 bp long locus. Using these conditions, we generated 7.3 M unlinked SNPs (the same number as the subsampled, unlinked empirical SNPs).

- The Atlantic Ocean vs. the Mediterranean Sea contrast (n = 67 vs n = 7 diploid samples)

Genome-wide $F_{ST}$ had been estimated to be 0.056, resulting in estimates of $T = 0.02885$ and $t = 176,572$ generations. Taking $T/2$ as the measure of time since the split, we simulated 7,349,210 SNPs:

ms 148 7349210 -t 18.57 -I 2 134 14 -ej 0.01443 2 1 -s 1 > simulated.out

We here specified to sample 148 chromosomes from the population (two times the n = 74 sample size), repeat the simulation 7,349,210 times, use the scaled population mutation rate ("-t 18.57"), sample two populations of 134 and 14 chromosomes each ("-I 2 134 14"), model a join between the two populations at generation time 0.01443 ("-ej 0.01443") and export one SNP per simulated locus ("-s 1").

- The North Eastern North Atlantic Ocean vs. the South Western North Atlantic Ocean (n = 47 vs. n = 20 diploid samples)

Genome-wide $F_{ST}$ had been estimated to be 0.0168 in this contrast, resulting in estimates of $T = 0.00825$ and $t = 51,752$ generations. We adjusted the number simulations to match the number of segregating sites in this sample and ran the matching simulation using the same approach as above:

ms 134 7211757 -t 18.57 -I 2 94 40 -ej $VAL 2 1 -s 1 > simulated.out

We converted the output from the ms simulations into allele counts and computed per-SNP $F_{ST}$ using the Weir-Cockerham estimator as above. We then compared the simulated $F_{ST}$-spectra to the observed (binning the SNPs in $F_{ST}$-bins of 0.1), in order to test for excess divergence compared to expectation under neutrality, which could be taken as evidence for natural selection.

**Signatures of selective sweeps.** For our two major pairwise contrasts (i: SW vs NE North Atlantic Ocean; ii: Atlantic Ocean vs Mediterranean Sea), we used the cross-population XP-nSL test in selscan v1.3.0[198] to scan for signatures of extended haplotypes in one group relative to the other. Such patterns could result from local selective sweeps through natural selection on an adaptive variant that reduces linked variation only in the focal population but not in the other, which is not subject to the selection pressure. XP-nSL can detect signatures of both hard and soft sweeps and help pinpoint candidate loci for ecological adaptation. It is conceptually similar to the XP-EHH test but does not require a genetic map or insight into recombination rates.

For each contrast, we first converted the VCF files to compressed TPED files. To limit the number of files on disk at a time, we then implemented a small daemon that generated one TPED subset per population and scaffold/contig and executed selscan:

selscan --xpnsl --threads 20 --trunc-ok --tped-ref <seq_$N.ref_population.tped > --tped <seq_$N.other_population.tped > --out <results_out/seq_$N.out>

For the first contrast, the NE North Atlantic Ocean sample (n = 47) was taken as the reference population and the SW North Atlantic Ocean sample as the other population (n = 20). For the second contrast, the Atlantic Ocean population (n = 67) was taken as the reference population and the Mediterranean sample as the other population (n = 7). In this scenario, negative XP-nSL scores for SNPs indicate extended haplotypes in the reference population, while positive scores are associated with the other population. We normalized the results with the selscan command "norm" against the genome-wide empirical background, such that extreme XP-nSL scores would be those less than -2 (indicating sweeps in the reference population) or more than 2 (indicating sweeps in the other population). We kept per-SNP XP-nSL scores and also computed the average across windows 1,000 bp.

**Enrichment analyses.** Genes were ranked from high to low exon-wide $F_{ST}$ and analyzed for enriched Biological Process gene ontologies using Flybase *Drosophila* homologues in GOrilla[161]. Enrichments, *p*-values and FDR-corrected q-values correcting multiple testing were computed by GOrilla. We retained all reported GO:s, which had FDRs q-values of about 0.1 or less.

The frequencies and enrichment of genes with ontologies associated with eye function were compared between $F_{ST}$-outliers and non-outliers. First, a standard two-tailed $\chi^2$-test was performed to test for a statistically significant difference between the observed and expected frequencies of such genes between outliers (0.1% $F_{ST}$-percentile) and non-outliers. We then assessed genes linked to expanded gene families. Among expanded genes, we compared outlier genes associated with eye function against those not associated with eye function (1% $F_{ST}$-percentile) vs. their frequencies in non-outlier genes. Conversely, among eye function genes, we compared genes associated with gene family expansion against those not associated with expansion. These tests were performed using the online GraphPad Software tool: https://www.graphpad.com/quickcalcs/contingency2/.

Tests and results are provided in Supplementary Data 10.

**Estimation of haplotype ages.** We estimated the ages of minor alleles on the divergent and putatively selected haplotypes to learn how long they may have been segregating in the Northern krill. For this, we used the nonparametric Genealogical Estimation of Variant Age (GEVA) coalescent method to estimate the time to the most recent common ancestor (TMRCA) between alleles[46]. GEVA uses both information about mutation and recombination rates to model TMRCA between ancestral and derived alleles, but does require a priori assumptions about demographic history.

Recombination rates were not known in krill from before. We therefore used iSMC v0.0.23[199] to estimate a genome-average recombination rate from the reference specimen alone, as this tool has been shown to provide robust estimates even from single diploid samples. iSMC uses the coalescent with recombination to infer both recombination rates and aspects of demographic history from heterozygous genotypes.

For estimation of recombination rate, we used genotypes from the reference specimen along 650 scaffolds longer than 500 kb and that had 60% or more accessible bases. We provided a VCF and a matching genome mask file. We left most settings at the defaults in the parameter file and used five rho categories for spatially heterogeneous recombination along sequences ("number_rho_categories = 5"), tolerance for numerical optimisation at 1e⁻⁴ ("function_tolerance = 1e-4"), a window size for decoding of 1 Mb (instead of the default 3 Mb, "fragment_size = 1000000") and 40 threads ("number_threads = 40"). The program was run:

ismc params=1.merged_contigs.bpp 2 > &1 | tee 1.merged_contigs.run.log

It infers the population recombination rate ρ, where:

$$\rho = 4 * Ne * r \qquad (7)$$

$N_e$ is the effective population size and r is the recombination rate per base pair. Our analyses gave a ρ of 0.013. For the reference specimen and set of 652 sequences, we estimated $\theta_W$ to 1.1% and $N_e$ accordingly to 1.02 M. We thus computed *r* to be 3.2e⁻⁰⁹ per bp or 0.32 cM/Mb across the 652 scaffolds, which we took as the genome-wide average.

We next prepared data for analysis in GEVA. We sought to analyze the age of variants on gene-haplotypes that diverge between Atlantic and Mediterranean ("at vs. me"; n = 660 genes) samples or between SW and NE Atlantic samples ("we vs. ea"; n = 34 genes) (our two major contrasts in these analyses). GEVA is designed to compute ages of derived alleles, which should be set as the ALT allele in re-coded VCF files. In our case, the ancestral and derived alleles were not known as we had not aligned outgroup sequences to the krill genome. We therefore instead re-coded the data assuming that the minor allele was derived (i.e. ALT) and the major allele was ancestral (i.e. REF) using a custom Perl script. For each contrast, we generated two sets of data, one set with the ALT allele taken as the minor allele in the first group (e.g. "at") and the other set with the ALT allele taken as the minor allele in the second group (e.g. "me"), respectively.

We converted the re-coded VCFs into GEVAs binary format (one file per contig/scaffold containing a gene of interest), specifying the recombination rate:

geva_v1beta -t 2 --vcf <data.vcf > --out <data.out > --rec 3.2e-09

We then executed the program, specifying $N_e$ and mutation rates and using the program-provided Hidden-Markov files:

geva_v1beta -t 2 --Ne 1530000 --mut 2.64e-09 --hmm hmm_initial_probs.txt hmm_emission_probs.txt

Allele ages of focal variants are estimated from a composite posterior distribution and saved in *.sites.txt output files. Each variant has an age based on a mutation clock (M), recombination clock (R) or joint clock (J). For all variants inside the gene coordinates of each gene, we collected the joint clock estimate and computed age distributions across all variants.

Results of these analyses are provided in Fig. 3.

## Assessment of molecular evolution in *nrf-6* and the topology of its encoded protein

In our scans for signatures of selection in the Northern krill, a homologue of the *nose resistant to fluoxetine protein 6* (*nrf-6*) gene encoding the NRF-6 protein was top-ranked for high $F_{ST}$ across its exons. To further learn about how selection may have acted on its gene and corresponding protein, we performed a comparative scan for positive selection between haplotypes. We overlaid the SNP variants detected in the Mediterranean samples on the reference (Atlantic) *nrf-6* CDS sequence and aligned them together with the homologous KrillDB2 sequence for the Antarctic krill *E. superba* (accession: ESS142994) and the decapod *P. vannamei* sequence (NCBI accession: QCYY01000544) using MAFFT. We then produced a 4-way phylogenetic tree with FastTree and analysed the dataset (alignment+tree) in PAML under a free-ratio codon model to infer per-branch *dN/dS* ratios[200] and test for elevated *dN/dS* on the Mediterranean haplotype of the gene.

We predicted its protein structure with Alphafold2, using the same approach as implemented for the DNA methylation genes (see above). The *nrf-6* gene encodes a transmembrane acyltransferase that assists in lipid transportation[47]. To predict the intracellular, transmembrane and extracellular regions of the protein and map the distribution of non-synonymous and synonymous variants along the regions, we used the TOPCONS server[201] and PPM v3.0 (positioning of proteins in membranes)[202]. We used SignalP v6.0[203] to predict the signal peptide. Protein figures were rendered using PyMOL[204] and

ChimeraX[205]. The coordinates reported by TOPCONS were then used to annotate the corresponding exons and locations of the detected SNPs.

Results of these analyses are provided in Fig. 4 and Supplementary Fig. 24.

### Inclusion & Ethics

This research was carried out collaboratively by an international and diverse team of researchers at different career-stages from local academic institutes located around the North Atlantic Ocean to Mediterranean Sea study area. The research was conceived by A.W., who had the leading role and built a network of collaborators to collect materials, sequence and analyze data and report results. As this biodiversity research neither involved CITES species, "higher invertebrates", animal testing or biorisks, no local ethics review committees, animal welfare, environmental protection and biorisk-related regulations were consulted (other than regulations regarding shipping and imports of samples into Sweden). Where relevant, our manuscript cites peer-review research of regional biodiversity patterns of krill or other animals. The research did not involve economical, cultural or social aspects or benefit sharing that would warrant review and engagement of local legal or stakeholder organizations. Biological reference material is deposited in the LIB Biobank at Museum Koenig Bonn, genetic sequence data is made publically available in the European Nucleotide Archive and SciLifeLab Data Repository and open source code is available on Github, to the benefit of all.

### Reporting summary

Further information on research design is available in the Nature Portfolio Reporting Summary linked to this article.

## Data availability

The sequence data generated in this study have been deposited in the public European Nucleotide Archive (ENA) database under accession code PRJEB61785. The processed data and results are available at the SciLifeLab Data Repository at https://doi.org/10.17044/scilifelab.c.6626216. The genome assembly is available in ENA and NCBI under accession code GCA_964058975.1 and SNP datasets are available under accession code PRJEB77093 in the European Variation Archive (EVA). Subsets of the data are provided in the Supplementary Information. Source data is provided as a Source Data file. This study made use data from the following public databases: AlphaFold Protein Structure Database https://alphafold.ebi.ac.uk/; Climate Reanalyzer https://climatereanalyzer.org/; Dfam (Dfam_3.5) https://dfam.org/home; Egg-NOG (v5.0) http://eggnog5.embl.de/; FlyBase database (release FB2021_01) https://flybase.org/; GOrilla https://cbl-gorilla.cs.technion.ac.il/; GyDB2 https://gydb.org; HomeoDB https://homeodb.zoo.ox.ac.uk/; KrillDB2 https://krilldb2.bio.unipd.it/; MITOS2 http://mitos.bioinf.uni-leipzig.de/; NCBI Conserved Domain Database (CDD) https://www.ncbi.nlm.nih.gov/Structure/cdd/wrpsb.cgi; NCBI Genome Database https://www.ncbi.nlm.nih.gov/genome/; NCBI RefSeq database (release 204) https://www.ncbi.nlm.nih.gov/refseq/; OrthoDB (v10.1) https://www.orthodb.org/; Pfam (release 34.0) http://pfam.xfam.org/; Repbase (RepBaseRepeatMaskerEdition 20181026) https://www.girinst.org/server/RepBase/; REXdb http://repeatexplorer.org/?page_id=918; ShinyGO 0.77 http://bioinformatics.sdstate.edu/go77/; SILVA rRNA database project (release 132) https://www.arb-silva.de/; The SWISS-MODEL Repository https://swissmodel.expasy.org/; TOPCONS https://topcons.cbr.su.se/; UniProtKB/Swiss-Prot https://www.uniprot.org/. Biological tissue from the reference specimen tissue is available in the LIB Biobank at Museum Koenig Bonn under accession ZFMK-TIS-82493. Three additional specimens are deposited under accessions ZFMK-TIS-82494 through ZFMK-TIS-82496. Source data are provided with this paper.

## Code availability

Public code is available at https://github.com/NBISweden/genecovr and https://github.com/andreaswallberg/Ecological-Genomics-Northern-Krill. A copy of the Github repositories is available on Zenodo: https://zenodo.org/doi/10.5281/zenodo.10827407.

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

## Acknowledgements

We thank Stéphane Plourde, Geneviève Perrin, Jon Rønning, Monica B. Martinussen and Katharina Michael for providing samples, the staff at Kristineberg Center for Marine Research and Innovation, Zhen Li for constructive discussion and Jessica Heinze, Sarah Demirkale and Ylva Jondelius for lab assistance. The computations were enabled project SNIC 2022/5-472 provided by the National Academic Infrastructure for Supercomputing in Sweden (NAISS) and the Swedish National Infrastructure for Computing (SNIC) at UPPMAX and the PDC Center for High Performance Computing partially funded by the Swedish Research Council through grant agreements no. 2022-06725 and no. 2018-05973. This research was supported by a Future research leaders grant awarded by the Swedish Research Council Formas 2017-00413 (A.W.) and by NSF OCE grants 1316040 and 1948162 (L.B.B.). PU was supported by the Knut and Alice Wallenberg Foundation as part of the National Bioinformatics Infrastructure Sweden at SciLifeLab, grant id KAW 2017.0003.

## Author contributions

Conceptualization: A.W. Data curation: A.W., P.U., I.B. Formal analysis: A.W., P.U., M.L., I.B. Funding acquisition: A.W. Investigation: A.W., A.O., A.P., O.W.; Methodology: A.W., P.U., A.P.; Project administration: A.W., O.V.P.; Resources: A.W., O.V.P., E.E., A.G., H.G., J.C., L.B.B.; Software: A.W., P.U.; Supervision: A.W.; Visualization: A.W., P.U., M.L.; Writing – original draft: A.W., P.U.; Writing – review & editing: A.W., P.U., B.M., L.B.B., E.E., A.G., H.G., J.C., C.P.

## Funding

## Competing interests

The authors declare no competing interests.

## Additional information

¹Department of Cell and Molecular Biology, National Bioinformatics Infrastructure Sweden, Science for Life Laboratory, Uppsala University, Uppsala, Sweden. ²Department of Medical Biochemistry and Microbiology, Uppsala University, Husargatan 3, 751 23, Uppsala, Sweden. ³Department of Pharmaceutical Biosciences, Uppsala University, Uppsala, Sweden. ⁴Uppsala Genome Center, Department of Immunology, Genetics and Pathology, Uppsala University, National Genomics Infrastructure hosted by SciLifeLab, Uppsala, Sweden. ⁵Biology Department, University of Padova, Padova, Italy. ⁶Marine and Freshwater Research Institute, Pelagic Division, Reykjavik, Iceland. ⁷Department of Biological Sciences, University of Bergen, Bergen, Norway. ⁸Center for Macroecology, Evolution and Climate Globe Institute, University of Copenhagen, Copenhagen, Denmark. ⁹Instituto de Ciencias del Mar (ICM-CSIC), Barcelona, Spain. ¹⁰Bermuda Institute of Ocean Sciences, Arizona State University, St. George's, Bermuda. ¹¹Institute of Marine Research (IMR), Bergen, Norway. ¹²Section Polar Biological Oceanography, Alfred Wegener Institute Helmholtz Centre for Polar and Marine Research, Bremerhaven, Germany. ¹³Institute for Chemistry and Biology of the Marine Environment, Carlvon Ossietzky University of Oldenburg, Oldenburg, Germany. ¹⁴Helmholtz Institute for Functional Marine Biodiversity (HIFMB), University of Oldenburg, Oldenburg, Germany. ✉e-mail: andreas.wallberg@imbim.uu.se

