## [Peer Review File · Nature Communications]

Ecological genomics in the Northern krill uncovers loci for local adaptation across ocean basinsREVIEWERS' COMMENTS

Reviewer #1 (Remarks to the Author):

- What are the noteworthy results?

This is a notable addition to our understanding of the biology of Northern krill, providing a number of insights into the evolution and adaptive potential of this key species in the pelagic community of the North Atlantic, Mediterranean and Arctic seas. The paper examined genome scale variation in specimens taken from most of the extremities of the distributional range of this species. Although there was homogenising gene flow, there was also evidence of adaptive divergence across a number of genes, principally involved in light perception, circadian regulation and moulting amongst others. A lipid transporter gene was found to exhibit a high degree of variation indicating local adaptation – the authors speculated this may allow the species to respond to the early spring conditions in the Mediterranean. There was also a number of interesting insights into lineage and previous population size although some results were equivocal.

- Will the work be of significance to the field and related fields? How does it compare to the established literature? If the work is not original, please provide relevant references.

I am not a molecular biology specialist although have a strong background in the biology and ecology of Northern krill. I found the paper to be adequately written in terms of drawing out the significance of the highly technical results on various genomic analyses. The fact that variation in the molecular results related to features known to be important in the biology of euphausiids, such as photoreception, moulting and thermal tolerance, added a high level of significance to the findings and contributes to a new level of understanding on the major drivers and limitations to euphausiid behaviour and population ecology. The work is highly original and makes a major step forward in our understanding of the molecular biology of this species.

- Does the work support the conclusions and claims, or is additional evidence needed?

Although my understanding of the molecular biology aspects of this work has limitations, I found the work to be thoroughly implemented and rigorous. The conclusions are consistent with the results presented throughout the work, with some caveats (see below).

- Are there any flaws in the data analysis, interpretation and conclusions? Do these prohibit publication or require revision?

I do not disagree with the conclusions but will point out to the authors that maybe further explanation is required as to why effective population size (NE) appears to be such an underestimate of the likely true population size (line 196) – although they provide some explanatory text (lines 197-202), they do not fully explore to what degree this error calls associated analyses into question.

Fig 2 cii – this is hard to interpret without a colour code – and why are the blocks for Spain without variation (please include an explanation in the legend)

Line 331 – from a biological perspective, it is not clear why moulting in euphausiids is so frequent compared to other crustaceans. One potential driver is to reduce external parasitism causing drag, which is worth mentioning here (Tarling, Geraint A., and J. Cuzin-Roudy. "External parasite infestation depends on moult-frequency and age in Antarctic krill (*Euphausia superba*)." *Polar Biology* 31 (2008): 121-13)

Line 368 – Velsch et al 1994 found that Northern krill had a 23 h circadian rhythm in constantly dark conditions (Velsch, J. P., & Champalbert, G. (1994). Rytmes d'activité natatoire chez *Meganyctiphanes norvegica* (Crustacea, euphausiacea). *Comptes rendus de l'Académie des*

sciences. Série 3, Sciences de la vie, 317(9), 857-862.)

Line 383 – it is not made clear why Scandinavian stocks in particular serve as a source of genetic diversity – this argument needs to be elaborated on much more clearly within the main body of the manuscript to be a concluding remark here

Line 395 – it would be helpful for the reader if the authors could specify here which candidate genes to focus on for such monitoring

- Is the methodology sound? Does the work meet the expected standards in your field?

I am not qualified to comment on all aspect of the molecular biological methods. I found the annotation of genes and their relationship to biological and ecology function to be sound

- Is there enough detail provided in the methods for the work to be reproduced?

The manuscript is thoroughly written and supported by extensive Supplementary material

Reviewer #2 (Remarks to the Author):

The new availability of a reference genome for the euphausiid, *Meganyctiphanes norvegica*, is an extremely useful resource for the research community, including oceanographers and molecular geneticists. The krill species has been a frequent focus of genetic, genomic, and transcriptomic studies for exactly the reasons the authors describe: the species' successful occupation and adaptation to an exceptional range of environmental / oceanographic conditions.

This study is a significant contribution and major step forward using long-read sequencing to produce a new reference genome, which is both exceptionally large and uniquely structured, and has drawn the attention of the research community for many years.

New evidence is reported regarding several persistent questions, including the role of present-day migration (i.e., dispersal) and natural selection versus demographic history in determining patterns of population genetic/genomic diversity and structure of this unique species.

The detailed analysis and whole-genome population resequencing, transcriptomic and epigenetic analyses provide solid evidence for novel conclusions about likely patterns of adaptation, evolution, and future responses to climate change. Of particular interest is identification and analysis of "gene families" that may play significant roles in the successful adaptation of the species to diverse environmental conditions across the geographic range.

The methodology is sound and state-of-the-art, meeting and exceeding standards for comparative genomic/transcriptomic analysis of non-model species. The methods are described in sufficient detail with necessary terminology (albeit with inevitable use of jargon and acronyms) to allow researchers in the field to repeat the analyses as desired, while still providing understandable summaries to non-specialists.

In summary, this is an outstanding and invaluable contribution to a fascinating question in the field of marine molecular evolution, focused on a zooplankton species that has captured the attention of researchers for many years.

RESPONSE TO REVIEWERS' COMMENTS

Reviewer #1 (Remarks to the Author):

- *What are the noteworthy results?*

This is a notable addition to our understanding of the biology of Northern krill, providing a number of insights into the evolution and adaptive potential of this key species in the pelagic community of the North Atlantic, Mediterranean and Arctic seas. The paper examined genome scale variation in specimens taken from most of the extremities of the distributional range of this species. Although there was homogenising gene flow, there was also evidence of adaptive divergence across a number of genes, principally involved in light perception, circadian regulation and moulting amongst others. A lipid transporter gene was found to exhibit a high degree of variation indicating local adaptation – the authors speculated this may allow the species to respond to the early spring conditions in the Mediterranean. There was also a number of interesting insights into lineage and previous population size although some results were equivocal.

This is a good summary of the work and we hope to clear up the equivocal results below.

- *Will the work be of significance to the field and related fields? How does it compare to the established literature? If the work is not original, please provide relevant references.*

I am not a molecular biology specialist although have a strong background in the biology and ecology of Northern krill. I found the paper to be adequately written in terms of drawing out the significance of the highly technical results on various genomic analyses. The fact that variation in the molecular results related to features known to be important in the biology of euphausiids, such as photoreception, moulting and thermal tolerance, added a high level of significance to the findings and contributes to a new level of understanding on the major drivers and limitations to euphausiid behaviour and population ecology. The work is highly original and makes a major step forward in our understanding of the molecular biology of this species.

Thank you! We appreciate this analysis and view.

- *Does the work support the conclusions and claims, or is additional evidence needed?*

Although my understanding of the molecular biology aspects of this work has limitations, I found the work to be thoroughly implemented and rigorous. The conclusions are consistent with the results presented throughout the work, with some caveats (see below).

Again, we appreciate this opinion and hope to resolve the caveats below.

- *Are there any flaws in the data analysis, interpretation and conclusions? Do these prohibit publication or require revision?*

I do not disagree with the conclusions but will point out to the authors that maybe further explanation is required as to why effective population size (N_e) appears to be such an underestimate of the likely true population size (line 196) – although they provide some explanatory text (lines 197-202), they do not fully explore to what degree this error calls associated analyses into question.

This is a fair point and we agree it deserves some more attention. While we would indeed expect large census population sizes (N_c) to be associated with large effective population sizes (N_e), it is well recognized that N_e as inferred from genetic data is often much smaller than N_c . In the neutralist population genetic framework, the effective population size is the size of an ideal population (that meets all Hardy-Weinberg assumptions) of reproductive individuals that over time would lose genetic variation through genetic drift at the same rate as the observed population. In this sense, N_e is typically a long-term measure and does not reflect the current-day number of reproductives. For example, N_e is often said to be $\sim 10,000$ in humans, many orders of magnitude lower than our current N_c . A common explanation for low N_e/N_c is that populations have expanded from severe bottlenecks characterized by low N_e and loss of genetic variation due to increased genetic drift. Under cycling conditions between large and small populations, long-term N_e equals the harmonic mean of N_e across generations, making it strongly influenced by the smallest population sizes over the time period. However, many other factors could contribute (see below). A low N_e/N_c ratio is common in marine species, see for example Hedgecock & Pudovkin 2011. Even the superabundant marine phytoplankton species *Emiliana huxleyi* appears to have a modest N_e , estimated to only $N_e \sim 2.7$ million (Krasovec et al. 2020).

We provided two inferences of N_e (Fig 2b):

1. a long-term inference under mutation-drift equilibrium for an idealized population, based on the mutation rate and observed levels of genetic variation in the population, which produces a single estimate of about 1.53 million under these analytical conditions
2. an inference using coalescent modeling with PSMC/MSMC, showing signatures of demographic fluctuations around the long-term estimate that is consistent with a recent expansion from a historical bottleneck, and which seem to increase beyond 2 million closer to the present.

For our analyses, we assume a mutation rate from snapping shrimp and use our empirical SNP dataset to estimate levels of genetic variation and infer N_e . We note that our estimates of N_e are comparable to those made from WGS data from the Antarctic krill by Shao et al. (2023) (~ 2 million) and those made by Choquet et al. (2023) for multiple species using more limited transcriptome data, including for *M. norvegica* (~ 1.8 million). Those works used a similar analytical framework as here. Shao et al. suggested that the ratio of N_e/N_c in the Antarctic krill could be as low as 5×10^{-9} ($2 \times 10^6 / 4 \times 10^{14}$). Assuming N_c of trillions also in the Northern krill, it could be very low in this species too. In theory, there is a risk that our analyses underestimate N_e , for example if the mutation rate used is too high and/or the observed levels of variation are too low. Moreover, simulations suggest that estimation of N_e in marine species with very large populations is difficult from limited point samples and likely to provide artificially low N_e/N_c ratios (Waples 2016). Lastly, because mutations accumulate slowly, neither the equilibrium approach or coalescent approach model effective population size close to the present, but provide long-term/historical estimates.

While we can not exclude the risk of biases in our estimates due to technical issues, we think (as Shao et al. did) that the low N_e/N_c ratio could also reflect biological processes. Charlesworth & Jensen (2022) list a number of potential factors that may shape genetic variation and influence its relationship with N_c : neutral processes such as mutational biases and a negative association between mutation rate and population size (the drift-barrier hypothesis). Additionally, the effects of demography could manifest from skewed distributions of offspring numbers (i.e. “sweepstakes reproductive success”), extinction and recolonization of local populations, and fluctuations in population size (e.g. bottlenecks). Moreover, background selection and recurrent selective sweeps could also drive genetic changes at linked sites within populations.

We observe clearly reduced levels of genetic variation up to about 50-100 kbp away from genes on average, which could be an indication of the effects of linked selection. Assuming ~30 kbp protein coding genes and an effect extending on average 75 kbp upstream and downstream of each gene locus, the effect of reduced genetic variation through linked selection could span almost 5 Gbp across the 19 Gbp genome. As stated above, we also detect signatures consistent with a population bottleneck.

The simple equation $N_E = \vartheta_w/4\mu$ could be unrealistic for this kind of species. It “assumes mutation-drift equilibrium, which is only reached after a number of generations of the order of N_e . As N_e increases, the assumption that the considered population has been devoid of bottlenecks and sweeps during the last N_e generations becomes less and less plausible” (Galtier & Rousselle 2020). With a 1-year or so generation time, it is possible that genetic variation in species like krill is influenced by many of the factors above, and that populations never reach the equilibrium levels of diversity before evolutionary and demographic processes perturb the variation. The challenges of accurately estimating effective population size from genetic variation and uncovering the processes and life history traits that influence it, and reconciling observed levels of genetic variation with census population sizes, are active research fields with many perspectives and methodologies (see for example Buffalo 2021 and Waples 2023).

To accommodate the reviewer’s concern we have considered the following:

1. Patterns of linkage disequilibrium (LD) and chromosomal segments of identity by descent (IBD) can also offer insights into population history. Long blocks of IBD suggest recent inbreeding or low N_e , and short blocks indicate large contemporary N_e . Likewise, the spectrum of the rates of decay of LD between SNPs at different genetic distances is influenced by how N_e has fluctuated in the recent past. While mutations accumulate slowly, recombination events are frequent and can be used to estimate recent N_e (e.g. Santiago et al. 2020).

We have now performed an inference of near-contemporary N_e using LD patterns with the program GONE, looking at three regional gene pools. This tool suggests a recent N_e of 4-5 million, consistent with our inferences of population expansion from a bottleneck (but still far below the census population size of trillions). We have written a short addition to the

methods section (lines 1829-1838; adding one reference), put the results into Supplementary Figure 19 and briefly expanded the results section:

Lines 191-192: “The rate of decay of linkage disequilibrium between SNPs in turn suggested N_e may recently have reached 4–5 million (Supplementary Fig. 19B)”

2. Population bottlenecks followed by an expansion may generate an excess of rare alleles, similar to selective sweeps. If populations have very different demographic histories, this could complicate interpretation of evolutionary processes. Our inferences of population mutation rate (ϑ_w) and nucleotide diversity (π) in each population (Supplementary Figure 19A) are very similar, suggesting they may have a shared recent evolutionary history.

We have now added per-population Tajima’s D to these analyses, which indicates a similar levels of excess of low-frequency variants compared to neutral expectation among all populations, for example due to similar demographic histories. Tajima’s D is slightly less negative in the Mediterranean sample, but the difference is small and we argue more samples are needed to distinguish between Atlantic and Mediterranean patterns. The results of these analyses have been added to Supplementary Figure 19A.

3. Although we estimate N_e in this work, not so many downstream analyses actually directly depend on the numerical estimate itself. The exception is the estimation of the recombination rate (using iSMC) followed by the estimation of the age distribution of minor alleles associated with divergent loci (using GEVA), which explicitly use estimates of N_e to derive a recombination frequency or sets of allele ages, respectively.

In the case of recombination rate, we estimate it to 0.32 cM/Mb using the re-arranged equation: $r = \rho / (4 * N_e)$, in which we inferred ρ to be 0.013 in iSMC and long-term N_e to be 1.02 million from the heterozygous genotypes in the deeply sequenced reference individual alone. Since this recombination rate was inferred from a single individual, we re-estimated ϑ_w and N_e for this analysis from that individual alone, in order to avoid using estimates not well-represented by that subset of data. In the case of GEVA, we used the global long-term estimate of N_e (1.53M).

In principle, a different N_e could be used, such as 4 million, which makes the recombination rate per base lower, i.e. $r = 0.085$ cM/Mb (an extremely low recombination rate). To test the effect that a larger N_e could have, we re-estimated the age distribution of minor alleles at divergent loci specifying both a lower recombination rate ($r = 0.085$ cM/Mb) and a larger effective population size ($N_e=4$ million) under the joint method in GEVA, combining estimates based on both mutation rate and recombination rate. Instead of the age of minor alleles being

centered around 1-2 million years (Fig 3e), they are inferred to be older, being centered around 4 million years (attached fig). However, since this very large N_e could be recent and does not appear to be supported by the data as representative of historical N_e , we argue that the long-term estimate as originally used is better suited for this analysis, and we choose to keep it as is. Moreover, our statement that most variants in the putatively adaptive loci appear to predate multiple glacial cycles appears to be true under both scenarios.

We acknowledge that estimating N_e is difficult in species like krill. We have applied several methods aimed to estimate N_e over different time periods. We do not think our estimates of much lower N_e than N_c are erroneous *per-se*. They could reflect evolutionary and demographic history and possibly also specific aspects of krill biology.

In addition to the edits above, we have added a section in the discussion bringing up why we may see low N_e/N_c ratio in krill, citing previously used krill references as well as the Charlesworth & Jensen 2022 review paper. We speculate that the mode of reproduction in krill could limit N_e due to skewed reproduction, but acknowledge that more research is needed to understand krill population dynamics. This text is written across lines 299-308.

We hope our response here and revised text provides enough context and explanation to be acceptable.

- Hedgecock & Pudovkin 2011: <https://www.ingentaconnect.com/content/umrsmas/bullmar/2011/00000087/00000004/art00013>
- Krasovec et al 2020: <https://academic.oup.com/gbe/article/12/7/1051/5869440>
- Waples 2016: <https://onlinelibrary.wiley.com/doi/10.1111/jfb.13143>
- Charlesworth & Jensen 2022: <https://academic.oup.com/gbe/article/14/7/evac096/6615374>
- Galtier & Rousselle 2020: <https://academic.oup.com/genetics/article/216/2/559/6066183>
- Buffalo 2021: <https://elifesciences.org/articles/67509>
- Waples 2023: <https://onlinelibrary.wiley.com/doi/full/10.1111/1755-0998.13879>
- Santiago et al 2020: <https://academic.oup.com/mbe/article/37/12/3642/5869049?login=true>

Fig 2 cii – this is hard to interpret without a colour code – and why are the blocks for Spain without variation (please include an explanation in the legend)

In Fig 2 cii, we show an admixture plot. In this analysis, a model-based framework is used to assign the observed genetic variation to one or more hypothetical ancestral pools or populations (given by the K number). This method simultaneously estimates and optimizes allele frequencies for ancestral populations and ancestry proportions for individuals from genotype data. It is rerun with different values of K, which is used to reveal genetic structure at different levels of resolution. Each bar reflects the overall proportions to which different hypothetical ancestors have contributed to the genetic fingerprint of an individual.

To perform this analysis, we ran the tool Admixture 50 times to analyze a subsample of LD-pruned SNPs, essentially producing 50 pseudoreplicate runs with different starting conditions for some parameters but using the same data. We then summarized the results using CLUMPAK, which identifies the major clustering modes across the 50 replicate runs (which individually may have had slightly more ambiguity in the ancestry assignment than the summary output).

We used the summary output for the figure and have avoided re-using the same color scheme for this plot that we used for other figures, such as Fig 2 ci & Fig 2 ciii, as these hypothetical ancestors do not necessarily reflect geographic structure. Samples from Spain stand out for having genetic variation that is comparably easily attributed to one of the ancestral populations. Their distinct genetic variation is also reflected in figures 2 ci and 2 ciii. We see no correspondingly strong genetic structuring between SW and NE Atlantic samples (consistent with the PCA overlap in i and low F_{ST} separating them in ii).

We have updated the legend to better explain the meaning of the admixture plot in Fig 2 cii and in the corresponding supporting figure (Supplementary Figure 20).

Line 331 – from a biological perspective, it is not clear why moulting in euphausiids is so frequent compared to other crustaceans. One potential driver is to reduce external parasitism causing drag, which is worth mentioning here (Tarling, Geraint A., and J. Cuzin-Roudy. "External parasite infestation depends on molt-frequency and age in Antarctic krill (*Euphausia superba*)." *Polar Biology* 31 (2008): 121-13)

We agree this deserves an explanation. We now state that the frequent molting in euphausiids is also a means to reduce load and drag from parasites and epibionts in the discussion. We quote both Tarling et al. as suggested and the chapter on parasites and disease by Gómez-Gutiérrez & Morales-Ávila in *Biology and Ecology of Antarctic Krill*, in order to link molting physiology to drag.

In the discussion we thus write:

“Molting is a crucial process in krill, being interlinked with growth and reproduction and a means to reduce load and drag from parasites and epibionts^{57,58} (lines 291-292)

Line 368 – Velsch et al 1994 found that Northern krill had a 23 h circadian rhythm in constantly dark conditions (Velsch, J. P., & Champalbert, G. (1994). Rytmes d'activité natatoire chez *Meganyctiphanes norvegica* (Crustacea, euphausiacea). Comptes rendus de l'Académie des sciences. Série 3, Sciences de la vie, 317(9), 857-862.)

Thank you for this remark. We have included this reference. Together with observations in the species we already brought up, this observation suggests that the short circadian rhythm is an ancestral feature in krill. We have edited this paragraph to include the reference and reflect that this is a shared trait among krill.

“*M. norvegica*, along with other krill species such as *E. superba* and *Thysanoessa inermis*, exhibit endogenous circadian rhythms (ECR) that cycle at rates faster than 24 hours in the absence of light^{70,71}, or respond to minute irradiance.” (lines 334-336)

In this revision, we are replacing the original Biscontin et al. 2017 reference with Biscontin et al. 2019, which addressed circadian rhythm in *E. superba* more comprehensively, at the transcriptome level.

Line 383 – it is not made clear why Scandinavian stocks in particular serve as a source of genetic diversity – this argument needs to be elaborated on much more clearly within the main body of the manuscript to be a concluding remark here

We appreciate the issue and have clarified these observations. When examining divergent loci, we found that Mediterranean haplotypes were relatively common in the Norwegian sample (mean frequency of 13%), while South-Western haplotypes were common in the Swedish sample (mean frequency of 17%). In contrast, the other Northern stocks do not harbor haplotypes from Southern regions to the same extent. These results are shown in Figs. 3diii and Figs. 3dvi, respectively. We have now described these results more clearly in the text:

“We found that Mediterranean haplotypes were comparably common in the Norwegian sample (mean frequency of 13%, Fig. 3diii), while South-Western haplotypes were frequent in the Swedish sample (mean frequency of 17%, Fig. 3dvi), suggesting these Scandinavian stocks in particular could contain genetic material that is otherwise rare at high latitudes.” (lines 229-232)

Line 395 – it would be helpful for the reader if the authors could specify here which candidate genes to focus on for such monitoring

We agree. Recommended markers for monitoring would be those that we observe to be geographically divergent and ecophysiologicaly relevant. As an example, we have now highlighted six genes linked to thermal stress or reproduction towards the end of the discussion.

We now write:

“The many candidate genes reported here can be used as biomarkers to diagnose and monitor change of adaptive variation in response to changing conditions, including *CCT8*, *HIP-R* and *Hsp83*

associated with thermal stress response and *nrf-6*, *Kr-h1* and *Hr3* involved in reproductive physiology, as well as those potentially coordinating photoperiodic regulation. These markers could also be used to monitor other species, in order to better forecast the future distributions of krill and estimate risks of the great many species that depend on them.” (lines 362-367)

• ***Is the methodology sound? Does the work meet the expected standards in your field?***

I am not qualified to comment on all aspect of the molecular biological methods. I found the annotation of genes and their relationship to biological and ecology function to be sound

• ***Is there enough detail provided in the methods for the work to be reproduced?***

The manuscript is thoroughly written and supported by extensive Supplementary material

Thank you for the careful and insightful review of this manuscript!

We thank the reviewer for the thorough and constructive review. We believe that the issues that they underscored have been directly relevant to central topics in our manuscript, and that addressing them have helped improve its quality. We hope that our replies and revisions have satisfyingly resolved each of the highlighted issues.

Reviewer #2 (Remarks to the Author):

The new availability of a reference genome for the euphausiid, *Meganyctiphanes norvegica*, is an extremely useful resource for the research community, including oceanographers and molecular geneticists. The krill species has been a frequent focus of genetic, genomic, and transcriptomic studies for exactly the reasons the authors describe: the species' successful occupation and adaptation to an exceptional range of environmental / oceanographic conditions.

This study is a significant contribution and major step forward using long-read sequencing to produce a new reference genome, which is both exceptionally large and uniquely structured, and has drawn the attention of the research community for many years.

New evidence is reported regarding several persistent questions, including the role of present-day migration (i.e., dispersal) and natural selection versus demographic history in determining patterns of population genetic/genomic diversity and structure of this unique species.

The detailed analysis and whole-genome population resequencing, transcriptomic and epigenetic analyses provide solid evidence for novel conclusions about likely patterns of adaptation, evolution, and future responses to climate change. Of particular interest is identification and analysis of "gene families" that may play significant roles in the successful adaptation of the species to diverse environmental conditions across the geographic range.

The methodology is sound and state-of-the-art, meeting and exceeding standards for comparative genomic/transcriptomic analysis of non-model species. The methods are described in sufficient detail with necessary terminology (albeit with inevitable use of jargon and acronyms) to allow researchers in the field to repeat the analyses as desired, while still providing understandable summaries to non-specialists.

In summary, this is an outstanding and invaluable contribution to a fascinating question in the field of marine molecular evolution, focused on a zooplankton species that has captured the attention of researchers for many years.

Thank you for the thoughtful and positive review of our manuscript!

We thank the reviewer for this feedback and hope they find our response and edits to have improved it further.

Other changes to the manuscript.

Below we list (A) updates, (B) corrections of technical issues and (C) editorial changes.

A. Updated analyses or reporting.

- 1. Results, Main figure 1f (update)** in the context of DNA methylation. We have enlarged Fig 1f and put the mean DNA methylation frequencies into the figure. We have also put these values into the legend for Supplementary Figure 17A, which has the same mean methylation frequencies presented as a bar plot. In Supp Fig 17A one bar (“non-repeats”) was misdrawn and has been corrected.

- 2. Results, Supplementary Fig 19D (update)** in the context of divergence. We originally performed a genetic isolation-by-distance test using the d_{XY} estimator against physical distance. We have realized such a test typically uses the F_{ST} estimator and specifically linearized F_{ST} (see Rousset 1997, 4,000+ citations). To improve our test, we have updated it to instead use this standard estimator for this context.

As in the original analysis, genetic distance only increases marginally with physical distance. The association between genetic and physical distance is weak when comparing North Atlantic Ocean vs. Mediterranean Sea populations ($R^2=0.47$; $P=0.09$) but strong when comparing North Atlantic Ocean populations ($R^2=0.69$; $P=3.346 \times 10^{-6}$), i.e. within the same ocean basin. These results match the F_{ST} -tree in Fig 2ciii well. The results have been reworded (line 197-199), a methods section has been written (lines 1900-1911) (which was originally missing) and values are reported in Supplementary Data 8.

- Rousset: <https://pubmed.ncbi.nlm.nih.gov/9093870/>

- 3. Results, Supplementary Data 8 (reporting)** in the context of divergent gene haplotypes. In the results, we address the haplotype distribution of 660 and 34 genes segregating among populations in two major contrasts (Fig. 3d-e), respectively. Information about the genes were missing. We have now added two gene lists to Supplementary Data 8 with per-gene information about the distribution of haplotypes in each population.

B. Corrections.

We have identified a few technical errors when preparing the revisions. We believe they do not meaningfully impact results or conclusions, but propose to correct them. We apologize for making these mistakes and hope the corrections are acceptable.

- 1. Results, in Supplementary Fig 17C** (formerly D) on DNA methylation. Bootstrap confidence intervals (CIs) for the P90 statistic (frequency of CpGs with >90% of reads methylated) were incorrectly estimated from mean methylation frequencies. We now use the correct P90 95% CIs, and longer whiskers to improve visibility. Methylation is structured as before (more methylation in genes and repeats), but has a little more variability within regions with this statistic.

- 2. Results, in Fig 3bi and Supplementary Fig 19C** in the context of divergence. These two subpanels were based on incomplete SNP data. Compared to the correct SNP dataset (~760 million SNPs) underlying all other analyses, simulations and results, this data spanned about 540 M SNPs and missed some heterozygous genotypes. The figures have been updated to be based on the complete dataset, as in the rest of the manuscript.

In Fig 3bi, the corrected plot is somewhat right-shifted: 1.2% instead of 0.9% of the genome has $F_{ST} > 0.4$ in the Atlantic vs Mediterranean contrast (at/me), and 0.08% vs. 0.05% in the Atlantic contrast (ea/we).

We have updated the main text in one place, from:

“Divergent regions ($F_{ST} > 0.4$) spanned <1% of the genome and were about 2× enriched for gene sequences and extended haplotypes”; to:

“Divergent regions ($F_{ST} > 0.4$) spanned ~1% or less of the genome and were about 2× enriched for gene sequences and extended haplotypes” (lines 215-216)

In Supplementary Fig 19C we provide an updated NJ-tree based on pairwise genetic distances. There are no meaningful changes to interrelationships among individuals. As before, the tree lacks structure and is star-like overall, with the Mediterranean samples forming the only recognizable group on a short separate branch.

3. **Results, in Figure 4** in the context of showing example candidate genes for adaptation. We have updated **Figure 4b*i***. The subpanel highlights divergence across an *rdgB* homolog, a gene involved in retinal photopigment metabolism. *rdgB* genes exist in several copies in the Northern krill, and more than one is divergent among populations. We accidentally highlighted the second most divergent gene (“REF_STRG_1_54916_XLOC_034835”) instead of the intended and most divergent one (“COM_SPAL_1_013733_XLOC_048553”). We have now replaced the subpanel with the intended gene.

In **4b*iii***, the x-axis scales were different for the upper F_{ST} plot and the lower XP-nSL plot for the *Acer* gene. The two plots have now been corrected to have the same range.

C. Editorial changes.

We have also made editorial changes such as correcting references or adding explanations to legends, or sample sizes, following instructions from the editors.

1. **In Results, references. Fig 1a**, we use a map with Sea Surface Temperatures from Climate Reanalyzer. As per their instructions:

To reference Climate Reanalyzer in a publication, please use the website citation below in addition to source citations for any datasets used:

Climate Reanalyzer (n.d.). [Title of specific page]. Climate Change Institute, University of Maine. Retrieved [Month Day, Year], from <https://climatoreanalyzer.org>

We have updated the reference as per these instructions. We have also followed the guidelines here: <https://psl.noaa.gov/data/gridded/data.noaa.oisst.v2.highres.html> to add Huang et al. (2021) to the reference list for the most recent OISST v2 citation, (adding it to the top of the methods section, which is slightly revised to accommodate this change) (lines 358-360). We have also clarified in the legend that circles indicate sample locations.

We have also replaced the original photo of a Northern krill (that was used with permission) with one recently taken by us.

2. **In Results, Fig 3**. We added the values of the respective F_{ST} percentiles indicated in **C i** and **ii** to the legend, as these were neither given in the text or the figure.
3. **In Results, Fig 4**. We added the values of the respective F_{ST} percentiles indicated in **A**, **B** and **C** to the legend, as these were neither given in the text or the figure.
4. **In Discussion, reference**. We have revised the former “Russel 1969” reference. This reference is for a book with the two main authors being Mauchline and Fisher (Russel is the editor). We have updated this reference to name the authors rather than the editor.
5. **In Results, Supplementary Fig 1**. Corrected a misalignment glitch in 1C about read lengths with not all N50 dots showing.
6. **In Results, Supplementary Fig 5**. A minor correction: “Assembly completeness is assessed by looking at the number of gene bodies covered to a certain extent at 75% identity (E).” It originally said 90% identity.
7. **In Results, Supplementary Fig 17**. A legend has been added to indicate the meaning of mean (black) and median (blue) circles in **A** (methylation rates in genomic regions). Sample size and explanation has been added to **D** (splice isoforms vs. DNA methylation):

“Only gene models supported by RNA-seq data and with at least one CpG site with

methylation data were included in the analysis (n=19,777 genes)".

The number of CpG sites were added to E. The section on sample size now says:

"Number of repeats per identity interval (left to right): n=7; n=91; n=1397; n=211. Number of CpG sites per identity interval (left to right): n=549; n=6,008; n=110,085; n=14,021."

- 8. In Results, Supplementary Fig 23.** Renumbered subpanels (from a, b ... to i, ii ...).
- 9. In Results, Supplementary Fig 25.** Added numbering to subpanels (i, ii ...).
- 10. In Results, Supplementary Data 4.** Minor changes to table titles/headers in file Supplementary Data 4 for clarification.
- 11. In Results, Supplementary Data 7.** This table originally contained accessions used to find and characterize DNA methylation genes through comparative analyses. It has now been expanded to also contain the source for the homeodomain sequences used to characterize Hox genes. The title and headers of the file have been updated accordingly.
- 12. In Methods.** Updates to database version specifications. Change of "NCBI Genbank" to "NCBI Genome Database" for the source of the bacterial genomes used to screen the assembly. Specified RefSeq release version (RefSeq release 204). KrillDB specified as KrillDB2. Dfam specified as Dfam_3.5. HOMEODB specified as HomeoDB2. Public databases used are now listed under 'Data Availability'.
- 13. Supplementary Notes.** The supplementary text sections about whole genome duplication and geographical regions of population samples are now written as Supplementary Note 1 and Supplementary Note 2.